

# Process-based three-layer synergistic optimal allocation model for complex water resource systems considering reclaimed water

**Jing Liu[1] Yue-Ping Xu[1*] Wei Zhang[2] Shiwu Wang[3] Siwei Chen[1]**

[1] Institute of Water Science and Engineering, College of Civil Engineering and Architecture, Zhejiang University, Hangzhou 310058, China

[2] College of Computer Science and Technology, Zhejiang University, Hangzhou 310058, China

[3] Zhejiang Institute of Hydraulics & Estuary, Hangzhou 310020, China

*Corresponding author*: Yue-Ping Xu, yuepingxu@zju.edu.cn

## Abstract

The increasing water demand due to human activities has aggravated water scarcity, and conflicts among stakeholders have increased the risk of unsustainable development. Ignoring the effects of trade-offs leads to misguided policy recommendations. This study highlights the concept of synergy among different aspects of water allocation process. A process-based three-layer synergistic optimal allocation (PTSOA) model is established to integrate the interests of stakeholders across subregions, decision levels and time steps while simultaneously coupling reclaimed water to establish



environmentally friendly solutions. A synergy degree index is constructed by applying
network analysis for optimization. PTSOA is applied in Yiwu City, Southeast China,
and is shown to improve the contradictions among different dimensionalities in a
complex system. Overall, $2.43\times10^7$~$3.95\times10^7$ m$^3$ of conventional water is saved, and
notable improvements in management are achieved. The application demonstrates the
efficiency and excellent performance of the PTSOA.
**Keywords** Three-layer optimization, water allocation, process, synergy, reclaimed
water

# 1. Introduction

Water scarcity has become one of the major impediments to the sustainable
development of cities (Yue et al., 2020). Emerging water scarcity concerns in cities are
associated with limited available water, severe water pollution and the relentlessly
growing demand for water as driven by industrial growth, population growth and higher
living standards; these factors have lead to intense competition for freshwater among
stakeholders of interest (Dai et al., 2018; Wu et al., 2023). However, the heterogeneous
distribution of water resources at both spatial and temporal scales is common in many
cities and results in water shortage risks and conflicts, which often require the
optimization of water resource allocation (Friesen et al. 2017). Moreover, some
satisfactory alternatives for individual stakeholders may result in negative externalities
on others. Therefore, it is critical to develop a synergistic optimal allocation model to





alleviate conflicts and ensure the security, efficiency, equality, eco-environmental
sustainability, and sustainable development of water systems simultaneously.

As equitable access to water resources is closely related to social stability, several

qualitative and indirect methods have been developed to assess water allocation
equality (D'Exelle et al. 2012). In cases with limited water resources, more water would
be allocated to users with better economic conditions to achieve more economic
benefits. Thus, stakeholders with poor economic status are ignored, resulting in
imbalanced development. Consequently, actions are often needed by local government
managers to avoid such situations. The Gini coefficient has been widely used to
evaluate equality and enhance the optimization of water allocation in water use sectors
(Xu et al. 2019; Hu et al. 2016; D'Exelle et al. 2012). However, it is unable to reflect
the dynamic interactions among objectives, i.e., how objectives interact with each other
and impact the equity of a system in cases with diverse alternative decisions. While, in
the perspective of coordinated alloaction, multiple goals are simultaneously considered
to avoid   negative effects as much as possible. Therefore, in addition to equity,
coordination should be considered in water allocation systems, and these two concepts
can be combined to promote systemic synergy. By identifying the dynamic interactions
among objectives, the internal mechanisms of a water system can be clarified, and
synergy can be achieved in cases with different potential decisions. It is also helpful to
identify the hurdles and opportunities associated with sustainable development for
cities and to establish specific action priorities for cities based on a comprehensive



understanding of the interactions among objectives. To address this knowledge gap, a
correlational network approach is applied in this study, and a synergy degree index is
presented to consider both the equity and coordination of water systems. Moreover,
systemic analysis is used to assess the level of coordination of complex objective
interactions in city water systems.

Network analysis, which has been widely used in studies of complex systems (Ball

et al., 2000; Saavedra et al., 2011; Bond, 2017), is a holistic approach for exploring the
characteristics of interactions among objectives. It provides clear visualization and
conceptualization of the interactions among variables to fully characterize those
interactions (Swain and Ranganathan, 2021). An array of network metrics (for example,
degree centrality, betweenness centrality, eigenvector centrality, closeness centrality,
and community) can be applied to quantify the importance of objectives or targets in an
interaction network (Zhou and Moinuddin, 2017) and reveal the strongly connected
pairs of goals or targets in the network (Allen et al., 2019). A key network metric in
such analysis is connectivity, which reflects the degree of coordination among different
objectives in a system; in synergy networks, high connectivity indicates that many
objectives can be achieved simultaneously and that the negative effects of interactions
are mild (Wu et al., 2022). Thus, to facilitate the discovery of high-quality decision
alternatives, alleviate negative conflicts among multiple utilities and inform decision
making, a synergy degree evaluation index is established and applied to the network
analysis of this study.



Due to the negative externalities of individual decisions, conflicts occur not only
across different users or objectives but also across hierarchical decision levels. Water
use contradictions and inconsistent decision making by multiple managers inevitably
results in trade-offs, including positive and negative water resource feedback in cases
with limited water availability (Wang et al., 2022). In practice, district administrators
allocate water to each sector in each subregion, and subregion managers then make use
decisions based on the allocated amount of water resources (Safari et al., 2014). Since
each decision maker places emphasis on different targets, feedback and coordination
among different decision makers are of great importance. Therefore, synergistic
hierarchical water allocation that achieves coordination among different decision
makers is imperative to avoid conflicts, save water and maintain social stability.
To address these hierarchical problems, bilevel programming (BLP) has been
widely used, wherein objectives at two hierarchical levels, namely, an upper level and
a lower level, are co-optimized (Zhang and Vesselinov, 2016; Jin et al., 2018). The
upper-level decision may be affected by the actions of the lower-level decision makers
(Arora and Gupta, 2009). Yue et al. (2020) formulated a bilevel programming (BLP)
framework to gain insight into the whole water allocation process with district
administrators and subregional farmers. Li et al. (2022) built a two-level model with
the overall interests of system managers at the top and the individual interests of water
supply departments at the bottom. The multilevel programming problem (MLPP) was
derived from the bilevel programming problem (BLPP) and is more applicable to real


world practices (Baky, 2014). However, limited studies have explored applying MLPP
(more than two levels) for water resource allocation, especially in cases with
unconventional water supplies.

To satisfy both long-term and short-term water needs and avoid unnecessary

administration costs and water resource use caused by a lack of coordination among
different allocation steps, temporally synergistic allocation and optimization are needed
(Haguma and Leconte, 2018). In annual water resource planning, the monthly
variability of hydrologic regimes and nonstationarity of the daily water demand must
be considered. As an alternative example of synergistic allocation at different time steps,
Vicuna et al. (2010) used a monthly nonlinear programming model and an annual
sampling stochastic dynamic programming (SSDP) model to establish a monthly
operating policy. Haguma et al. (2015) proposed an optimization approach with two
separate time steps following the nested model approach. Haguma and Leconte (2018)
constructed deterministic and stochastic optimization models with two time steps (intra-
annual and interannual) and two levels of inflow variability: seasonal and interannual.
The purpose of their short-time-step model was to derive aggregate performance
functions associated with potential long-time-step decisions in these studies. However,
short-term benefits should not be overlooked due to their appreciable impact on long-
term effects. Accordingly, synergistic allocation that enhances both long-term and
short-term allocations is of great importance for water resource management in cities.
However, optimizing the structure of a model to achieve maximized benefits and

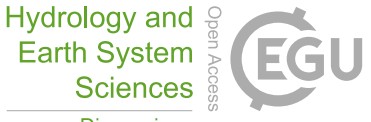

balancing the trade-offs among time steps are tasks that have rarely been studied. The
synergy among different time steps is addressed with a new innovative framework and
a corresponding algorithm in our study.

Most of the abovementioned traditional models are based on a benefit-oriented

mechanism, which leads to a high degree of satisfaction in high-benefit regions and
large water shortages in other regions. The existence of high-benefit regions in a city
during the allocation process often exacerbates regional disparities and heterogeneous
development. Moreover, spatial factors influence allocation results, especially when
there is spatial hierarchical heterogeneity among water resource allocation elements (Li
et al., 2022). It is thus appropriate to conceptualize water allocation problems in a
multistage framework that fully considers the interests of not only the regional authority
but also subregional managers (Yao et al., 2019). Hence, the synergy among subregions
must be considered to optimally allocate water resources. Ideally, the benefits of all
subregions should be integrated equally in the model, and the weights of
hyperparameters should be adjusted to best support flexible policies.

The optimal allocation of conventional and unconventional water resources also

significantly impacts water security and aquatic ecosystems. The reuse of reclaimed
water is beneficial for alleviating high water supply pressure on conventional water
resources and reducing the emission of pollutants. To effectively integrate conventional
and unconventional water resources, Yang et al. (2008) and Han et al. (2008) introduced
unconventional water resources as critical factors in water management. Avni et al.



(2013) investigated the mixing of unconventional water resources with other
conventional water sources to meet the magnesium requirements for drinking water and
irrigation water. Yu et al. (2017) developed a cost–benefit analysis-based utilization
model for externally transferred water and desalinated water. The allocation of both
conventional and unconventional water has been widely studied, but there remains a
lack of methods to guide the synergistic allocation of conventional and unconventional
water resources and embed reclaimed water supply systems in allocation schemes. The
overexploitation of conventional water resources is not conducive to the sustainable
development, while the extensive use of unconventional water could ultimately result
in high economic burden. To synergistically integrate conventional and unconventional
water resources and guide the coordinated allocation of these two types of water
resources, corresponding mechanisms must be implemented. As a result, our study aims
to couple the allocation of conventional water resources and unconventional water
resources to establish synergistic solutions.

In summary, as insufficient water supplies and increasing water demands intensify

competition for water resources and lead to conflicts among different stakeholders in
different dimensions, water allocation must be optimized in cities and regions to
achieve synergistic decision-making at various levels and time steps considering the
value of reclaimed water. Therefore, a new process-based three-layer synergistic
optimal allocation (PTSOA) model is developed here to generate numerous candidates
or Pareto solutions and identify several desirable decision alternatives. The synergy of


time and space optimization is achieved in the new model to avoid waste and promote
balanced spatial development. Furthermore, in the PTSOA model, reclaimed water is
used to replenish conventional water resources in water-scarce areas.

The remainder of this paper is organized as follows. The mathematical model is

formulated in Section 2. Section 3 gives a numerical example for Yiwu city to
demonstrate the effectiveness and efficiency of the proposed methods. The results are
shown in Section 4; different water allocation strategies under varying inflow
conditions are explored, and some policy implications are discussed. Section 5 presents
conclusions.

## 174    2. Modelling

With water resources becoming increasingly scarce, multidimensional synergistic
optimal allocation in a hierarchical system is crucial for ensuring sustainable
development in water-scarce cities. There are three dimensions of synergy in the
established allocation model, as shown in Fig. 1: process synergy, decision-level
synergy and time-scale synergy. The synergy of the process refers to synergistic water
allocation among the three stages throughout the whole allocation process to reduce
waste in bridging processes, which has rarely been considered. In the three stages, first,
the original water is released from reservoirs or diverted from external water transfer
projects to water works; then, the water stored in water works infrastructure is supplied
to different departments that need different types of water, including both conventional





and reclaimed water. Finally, the water is supplied to different users. Decision-level
synergy refers to synergistic water allocation considering the interests of decision
makers at different levels, namely, the city, water department and regional levels, to
coordinate solutions and avoid conflicts among decision makers. The city level
represents the overall interests of a city from the perspective of government, the water
supply department level represents the interests of water supply corporations, and the
regional level focuses on the comprehensive benefits of each region in the city and
mitigate development imbalance among regions. Optimal decision making at the
department level is constrained by the allocation results at the city level, and so on, and
the final solution should satisfy the needs of decision makers across all levels. The time
scale synergy involves the coordination of the daily configuration goal with the monthly
goal, the monthly goal with the yearly goal, and so on. Synergistic temporal allocation
can largely alleviate time conflicts during configuration operations, ensuring that all
configuration periods serve the same final configuration objectives to save water
resources and improve efficiency. However, time scale synergy mainly depends on
artificial operations rather than automated intelligent operations in practice. In-depth
exploration has yet to be demonstrated. Consequently, the PTSOA model is constructed
here to fully consider these three dimensionalities of synergy. The dimensionalities are
coupled this model to achieve the efficient maximization of comprehensive benefits at
all levels under the premise of saving water resources.


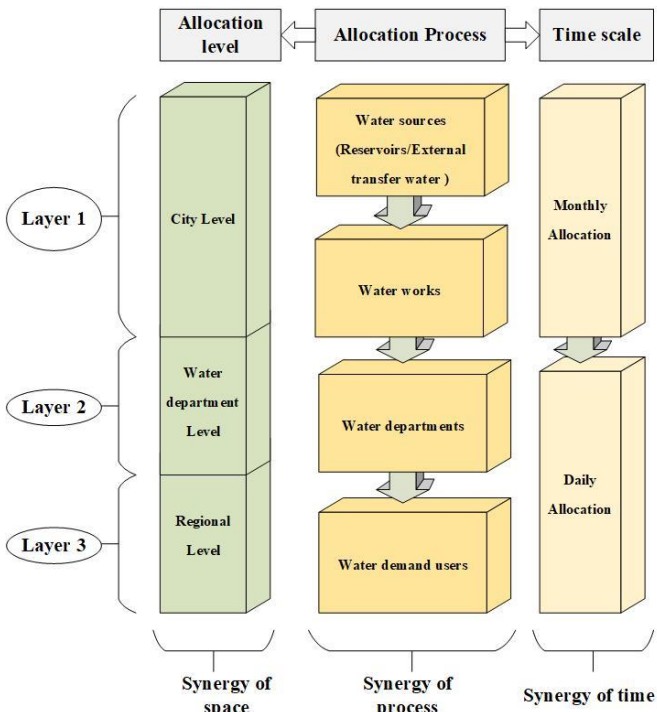


**Fig. 1. Conceptual map of the PTSOA model**


In water-scarce cities, using reclaimed water as an alternative water source is a
proven approach to efficiently improve the environment by reducing sewage discharge.
The quality of inland tributaries has deteriorated in many water-scarce cities due to
limited consideration of the water environment and the large-scale emission of
pollutants. Transferring reclaimed water and main river water to urban inland tributaries
for ecological water replenishment is a promising approach for improving the quality
of urban water environments and areas with water shortages. However, there has been
a lack of studies on the integration of reclaimed water reuse systems and inland water



distribution systems in allocation modelling. Therefore, in addition to saving water
resources and improving efficiency through multidimensional synergistic allocation,
the model encompasses reclaimed water reuse systems and ecological water
distribution systems for inland tributaries.

Finally, the PTSOA model is constructed to solve the multidimensional synergistic

allocation problem involving complex water resource networks that couple reclaimed
water reuse systems and inland ecological water distribution systems with multiple
sources, processes and regions to guarantee the sustainable development of water-
scarce cities. To select the most synergistic solution of the PTSOA model, a new
evaluation index named the total synergy index (TSI) is proposed to assess the synergy
degree of different decision alternatives. Furthermore, the network analysis method is
applied for the first time to analyse dynamic interactionsin water optimal allocation.
This method visually depicts the dynamic interactions and conflicts among different
subareas in a city, which is helpful for system managers to realize how the water
allocation scheme in one region influences that of other areas; consequently, more
reasonable and flexible measures are established based on dynamic regional
development targets. The detailed framework developed in this study is shown in Fig.

2.



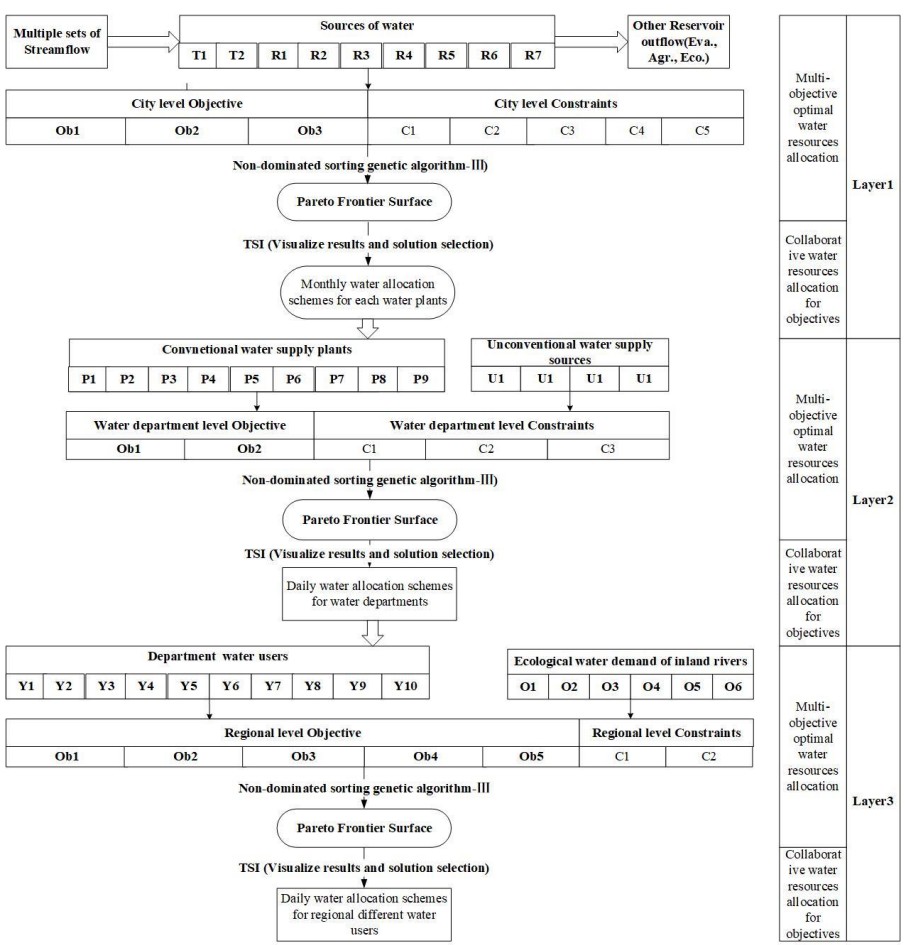


**Fig. 2. Framework of the PTSOA model**

## 2.1 First layer of the PTSOA decision-making process

Three dimensionalities of synergistic water resource allocation are coupled in the first
layer of the PTSOA model. The first stage of the process (original water is released
from reservoirs or external water transfer projects to water works) is optimized in the
first layer. This stage demonstrates a strong constraint effect on the later stages. To





satisfy the overall development goals of the city, the first-layer processes involve city-
level decision-making. The city manager focuses on the overall goal of the water
resource system in the city, which is the first and most important phase of the decision-
making process. The established allocation scheme highly influences decision makers
at other levels, and optimal allocation schemes at other levels must align with this
overall goal. Additionally, since water resource planning in most Chinese cities is based
on an annual planning period and monthly planning unit, the time step of the first layer
is set as months. Finally, the monthly decision alternatives for the volume of water
allocated from reservoirs to water works is obtained at the city decision level.

### 2.1.1 Objective functions

**Social objective function: Minimization of total water supply shortages**

The social objective function is established by the city manager to minimize the total

water supply shortages in a water system. The objective is established to sufficiently

meet the water demands of users in a water resourcees system. The water deficit is

considered, and this objective can reflect the ability of the water supply to meet the

water demand, as shown in Eqs. (1-3):

$$\min f_{11}(x) = D - S \tag{1}$$

$$D = \sum_{r=1}^{R}\sum_{t=1}^{T} D_r^t \tag{2}$$

$$S = \sum_{t=1}^{T}\sum_{i=1}^{I}\sum_{j=1}^{J} x_{ij}^t \alpha_{ij} + \sum_{t=1}^{T}\sum_{e=1}^{E}\sum_{j=1}^{J} x_{ej}^t \beta_{ej} \tag{3}$$





where $D\,(10^4\,\text{m}^3)$ is the total water demand of the system, $D_r^t\,(10^4\,\text{m}^3)$ is the water
demand of the $r$th subregion at $t$th time step, $r=1,2,...,R$ , $R$ is the total number of
subregions in the area, $t=1,2,...,T$, $T$ is the total number of months in the period, $S\,(10^4$
m³) is the total water supply of the water system for all waterworks, $x_{ij}^t\,(10^4\,\text{m}^3)$ is the
amount of water supplied from $i$th reservoir to the $j$th waterworks in the $t$th month of
the configuration period, $i=1,2,...I$, $I$ refers to the total number of reservoirs, $j=1,2,...J$,
$J$ is the number of total water works, $x_{ej}^t\,(10^4\,\text{m}^3)$ is the amount of water supplied from
the $e$th external transfer water source to the $j$th water works in the $t$th month of the
configuration period, $e=1,2,...,E$, $E$ is the total number of external transfer water
sources in the city, $\alpha_{ij}$ is the water supply relationship coefficient between the $i$th
reservoir and the $j$th water works, where 0 indicates no supply and 1 indicates a water
supply, and $\beta_{ej}$ is the water supply relationship coefficient between the $e$th external
transfer water source and the $j$th water works, where 0 indicates no supply and 1
indicates a water supply.
**Economic objective function: Maximization of the total water supply benefit**
A city manager operates a water allocation system to maximize the overall economic
benefit by establishing an economic objective function, as shown in Eqs. (4-7):
$$\max f_{12}(x) = B - C_{rs} - C_{es} \tag{4}$$
$$C_{rs} = k \times \sum_{t=1}^{T}\sum_{i=1}^{I}\sum_{j=1}^{J} x_{ij}^t \alpha_{ij} + \sum_{t=1}^{T}\sum_{i=1}^{I}\sum_{j=1}^{J}\left(x_{ij}^t \alpha_{ij} \times c_i\right) \tag{5}$$
$$C_{es} = m \times \sum_{t=1}^{T}\sum_{j=1}^{J}\sum_{e=1}^{E} x_{ej}^t \beta_{ej} + \sum_{t=1}^{T}\sum_{e=1}^{E}\left(n_e \times \sum_{j=1}^{J} x_{ej}^t \beta_{ej}\right) \tag{6}$$



$$B = \sum_{j=1}^{J} b_j \times \left( \sum_{t=1}^{T} \sum_{i=1}^{I} x_{ij}^t \alpha_{ij} + \sum_{t=1}^{T} \sum_{e=1}^{E} x_{ej}^t \beta_{ej} \right) \qquad (7)$$


282 The overall economic benefit is the difference between the total benefit and total

283 cost at the city level. In the equations, $B$ (Chinese yuan, shortened to yuan in the

284 following text) is the total direct water supply benefit (mainly considering the income

285 from water charges for the city). The total water supply cost consists of the reservoir

286 water supply cost $C_{rs}$ and the external water supply cost $C_{es}$; $k$ (yuan/m$^3$) denotes

287 the water resources fees paid to the government; $c_i$ (yuan/m$^3$) denotes the water fees

288 paid to the $i$th reservoir authority; $m$ (yuan/m$^3$) is the charge to an external

289 administrative district per unit of externally transferred water; $n_e$ (yuan/m$^3$) is the

290 charge associated with the $e$th external water source per unit of transferred water; and

291 $b_j$ (yuan/m$^3$) is the unit price of water supply revenue for the $j$th user.

292 **Sustainable development objective function: Maximization of the total amount of**

293 **reserved water in reservoirs**

294 In water-scarce cities, the problem of water scarcity is a serious challenge that prevents

295 sustainable allocation of water resources. A prominent feature of most water-scarce

296 cities is that water inflows are limited, and the fluctuations in water availability are

297 large. Therefore, to reduce the risk that the inflows in the next configuration period are

298 too short to meet the basic demand of the city, such that a sustainable development

299 objective function is developed. The sustainable development objective function seeks

300 to maximize the amount of water remaining in the reservoir at the end of a configuration





period to hedge against drought risk and guarantee water use in the next period, as
shown in Eqs. (8-10):

$$f_{13}(x) = \max \sum_{i=1}^{N} \left( V_i^{\max} - V_i \right) \times p(V_i^{\max} - V_i) \qquad (8)$$

$$p(V_i^{\max} - V_i) = \begin{cases} 2 \times V_i / V_i^{\max} & 0 < V_i < V_i^{\max}/2 \\ -2 \times V_i / V_i^{\max} + 2 & V_i \geq V_i^{\max}/2 \end{cases} \qquad (9)$$

$$V_i = \sum_{t=1}^{T} \left( R_{i,initial} + I_i^t + P_i^t - A_i^t - E_i^t - EP_i^t \right) - \sum_{t=1}^{T} \sum_{j=1}^{J} x_{ij}^t \alpha_{ij} \qquad (10)$$

where $V_i^{\max}$ ($10^4\,\mathrm{m^3}$) is the maximum allowable storage capacity of the $i$th water source,
which is expressed based on the limited storage capacity of a reservoir in the flood
season, and $V_i$ ($10^4\,\mathrm{m^3}$) is the water storage capacity of the $i$th water source at the end
of the scheduling period. As much water as possible but less than $V_i^{\max}$ is reserved.
However, a reserved water volume in the reservoir that is too high at the end of the
scheduling period may lead to considerable pressure on reservoirs to urgently release
water if a flood event is forecasted. The reserved water volume should be neither too
large nor too small. Thus, the benefits of reservoir retainment must be thoroughly
evaluated. Based on the characteristic of the benefit of residual water, we propose a
boundary benefit function $p(V_i^{\max} - V_i)$ for different reserved water volumes in a
reservoir. The benefit function is a piecewise function, and when $V_i$ is less than
$V_i^{\max}/2$, $p$ increases as $V_i$ increases. When $V_i$ is equal to or greater than $V_i^{\max}/2$,
$p$ decreases as $V_i$ increases. When $V_i = V_i^{\max}$, $p$ decreases to 0. $R_{i,initial}$ ($10^4\,\mathrm{m^3}$) is
the initial storage of the $i$th water source, $I_i^t$ ($10^4\,\mathrm{m^3}$) is the inflow of the $i$th water source





at the $t$th time step, $P_i^t$ ($10^4$ m³) is the precipitation associated with the $i$th water source
at the $t$th time step, $A_i^t$ ($10^4$ m³) and $E_i^t$ ($10^4$ m³) are the agricultural and ecological
water supplies associated with the $i$th water source at the $t$th time step, respectively, and
$EP_i^t$ ($10^4$ m³) is the evaporation from the $i$th water source at the $t$th time step.
## 2.1.2 Constraints
The layer includes six main constraints: the reservoir water supply constraint, water
demand constraint, reservoir storage constraint, water balance constraint, external water
transfer constraint, and nonnegative constraint.
**Reservoir water supply constraint**
The maximum water available to supply from an individual reservoir is determined by
the difference between the total input and total reservoir output. The inputs include
inflow and precipitation, and the outputs mainly involve agricultural and environmental
water supplies, evaporation, water supplied for waterworks and reservoir leakage loss.
All these factors directly affect the decision-making process and are incorporated into
the model building process as shown in Eqs. (11-15):
$$V_i^t \leq V_{i,\max}^t \tag{11}$$

$$V_i^t = \sum_{j=1}^{J} x_{ij}^t \alpha_{ij} \tag{12}$$

$$V_{i,\max}^t = \sum_{t=1}^{t-1} \left( R_{i,initial} + I_i^t + P_i^t - A_i^t - E_i^t - EP_i^t - \sum_{j=1}^{J} x_{ij}^t \alpha_{ij} - L_i^t \right) - V_{i,d} \tag{13}$$

$$EP_i^t = ep_i^t \times s_i^t / 1000 \tag{14}$$



$$V_i^t = \xi_i^t \times \left( R_i^{t-1} + R_i^t \right) \tag{15}$$

where $V_i^t$ ($10^4$ m³) denotes the total water supply from the $i$th reservoir at the $t$th time
step; $V_{i,\max}^t$ ($10^4$ m³) is the maximum water available to be supplied from the $i$th
reservoir at the $t$th time step; $ep_i^t$ (mm) is the water surface evaporation from the $i$th
reservoir in the $t$th month; $s_i^t$ (m²) is the monthly average surface area of the $i$th
reservoir in the $t$th month; $V_{i,d}$ ($10^4$ m³) is the dead storage of the $i$th reservoir; $L_i^t$
($10^4$ m³) is the reservoir leakage loss from the $i$th reservoir at the $t$th time step; $R_i^{t-1}$
($10^4$ m³) is the storage of the $i$th reservoir at the $t$-1th time step; $R_i^t$ ($10^4$ m³) is the
storage of the $i$th reservoir at the $t$th time step; and $\xi_i^t$ is the $t$th monthly leakage
coefficient for the $i$th reservoir.
**Water demand constraint**
The high-quality water demand of each subarea in a city should be satisfied in the water
allocation process. High-quality water in this model refers to water that satisfies the
relevant primary (surface water can be used for drinking after simple purification
treatment, such as filtration and disinfection) and secondary water quality requirements
(water is slightly polluted and can be used for drinking after routine purification
treatment, such as flocculation, precipitation, filtration, disinfection, and other
processes) according to the Chinese Standard (GB5749), as shown in Eq. (16):

$$0.8 \times D_r \le \sum_{t=1}^{T}\sum_{i=1}^{I}\sum_{j=1}^{Jr} x_{ij}^t \alpha_{ij} + \sum_{t=1}^{T}\sum_{e=1}^{E}\sum_{j=1}^{Jr} x_{ej}^t \beta_{ej} \le 1.2 \times D_r, r=1,2,...,R \tag{16}$$



where $D_r$ ($10^4$ m³) is the high-quality water demand in the $r$th subregion and there are
a total of $R$ subregions in the city. $Jr$ is the number of waterworks in the $r$th subregion.
To ensure that the water supply guarantee in each area is greater than 80%, the total
water supplied to every subarea is greater than 80% of its demand.
**Reservoir storage constraint**
$$R_i^T \leq V_{i,f} \tag{17}$$

$$R_i^T = \sum_{t=1}^{T} \left( R_{i,initial} + I_i^t + P_i^t - A_i^t - E_i^t - EP_i^{t1} - \sum_{j=1}^{J} x_{ij}^t \alpha_{ij} - V_i^t \right) \tag{18}$$

where $R_i^T$ ($10^4$ m³) is the storage of the $i$th reservoir at the end of the configuration
period and $V_{i,f}$ ($10^4$ m³) is the flood-limit storage capacity.
**Water balance constraint**
$$R_i^{t+1} = R_i^t + I_i^t + P_i^t - A_i^t - EP_i^t - E_i^t - V_i^{t-1} - \sum_{j=1}^{J} x_{ij}^t \tag{19}$$

**External transfer water constraint**
$$\sum_{t=1}^{T} \sum_{j=1}^{J} x_{ej}^t \beta_{ej} \leq E_{e,\max} \tag{20}$$

where $E_{e,\max}$ refers to the maximum water supply capacity of *an* external water source
over the whole configuration period.
**Nonnegative constraint**
$$x_{ij} \geq 0 \tag{21}$$



## 2.2 Second layer of the PTSOA decision-making model

Similarly, the second layer of the PTSOA model fuses all three dimensions of
synergistic water resource allocation mentioned previously. The second stage of the
process (the water stored in water works is supplied to different departments needing
water volumes of different quality) is optimized in the second layer. After city-level
decision-making, a conflict of interest inevitably occurs between traditional water
supply departments and unconventional water supply departments. Because
conventional and unconventional water supply departments compete for limited water
demand market shares, the stability of the water allocation system may be jeopardized
if excessive competition is not controlled. Thus, the second layer is implemented at the
department level. Decision-making at the department level seeks to guide the two water
supply departments to partake in benign competition and avoid conflicts to realize
synergy. In this case, the decision plan of the first layer in the hierarchy is followed.
Temporally, short-term allocation changes are needed as mentioned above; hence, the
time scale of the second layer is daily. Thus, the daily decision alternatives for the
volume of water allocated from water works to different water departments are obtained
to make relevant decisions.

### 2.2.1 Objective functions

**Conventional water supply department objective function: Minimization of the**



**total amount of water retained in water works**

The managers of conventional water supply departments strive to operate conventional water systems efficiently and achieve the most equitable water share possible. The amount of conventional water (of high quality) retained in a water works system is a crucial factor affecting the efficiency and benefits of conventional water supply departments. Therefore, the benefit of conventional water departments is established by minimizing the total amount of water retained in water works at the end of a configuration period, as shown in Eq. (22):

$$\min f_{21}(x) = W_L = \sum_{t=1}^{T}\sum_{i=1}^{I}\sum_{j=1}^{J} x_{ij}^{t}\alpha_{ij} + \sum_{t=1}^{T}\sum_{e=1}^{E}\sum_{j=1}^{J} x_{ej}^{t}\beta_{ej} - \sum_{t=1}^{T}\sum_{m=1}^{M}\sum_{j=1}^{J}\sum_{z=1}^{Z} q_{jz}^{t,m}\chi_{jz} \qquad (22)$$

where $W_L$ ($10^4$ m1) is the total amount of water retained in a water works system at the end of a configuration period; $q_{jz}^{t,m}$ ($10^4$ m³) is the water supply from the $j$th water works system to the $z$th water user on the $m$th day in the $t$th month in the period of configuration; $m=1,2,...,M$; and M is the total number of days in the $t$th month (28, 29, 30 or 31). Additionally, $z=1,2,...Z$, and Z is the total number of water users. $\chi_{jz}$ is the water supply relationship coefficient between the $j$th water work and the $z$th water user, where 0 indicates no supply and 1 indicates a water supply.

**Unconventional water supply objective function: Maximization of the amount of unconventional water supplied**

The reclaimed water reuse system and ecological water distribution system for inland tributaries are incorporated into the PTSOA model which are associated with





unconventional water supply departments. The managers of unconventional water
supply departments seek to supply as much unconventional water as possible to
promote their interests. Thus, the objective of unconventional water departments is
established to maximize the amount of unconventional water supplied. Unconventional
water mainly includes reclaimed water and river water, which is of low quality (i.e., not
meeting the quality standard mentioned in Sect. 2.1.2) and is mainly used for industrial
production, ecological water replenishment for inland rivers and municipal road
sprinkling.
Unconventional water departments operate reclaimed water reuse systems and
ecological water distribution systems to supply unconventional water, and the
associated equations are as follows in Eqs. (23-26):
$$\max f_{22}(x) = W_r + EW_r \tag{23}$$

$$W_r = \sum_{t=1}^{T}\sum_{n=1}^{N}\sum_{j=1}^{J} r_{nj}^{t}\, p(b_c, b_u)\,\theta_{nj} \tag{24}$$

$$p(b_c, b_u) = \frac{1}{3}\times\frac{b_c}{b_u} - \frac{2}{3} \tag{25}$$

$$EW_r = \sum_{t=1}^{T}\sum_{n=1}^{N}\sum_{z=1}^{Z} r_{nz}\,\theta_{nz} \tag{26}$$

where  $W_r$  ($10^4$ m³) is the total amount of reclaimed water supplied for all water users;
$EW_r$  ($10^4$ m³) is the total amount of river water supplied to maintain ecological flows
in inland tributaries;  $r_{nj}^{t}$  ($10^4$ m³) is the amount of water supplied to the $j$th user from
the $n$th reclaimed water source at the $t$th time step; $n = 1,...,N$; $N$ is the total number of





reclaimed water sources; $p\left(b_c, b_u\right)$ is a function expressing the willingness of residents
to use reclaimed water, $b_c$ (yuan/$10^4$ m$^3$) is the price per unit of conventional water;
$b_u$ (yuan/$10^4$ m$^3$) is the price per unit of unconventional water; and $\theta_{nj}$ is the water
supply relationship between the $n$th reclaimed water source and the $j$th user. In this case,
$\theta_{nj} = 1$ indicates a water supply relationship, and $\theta_{nj} = 0$ indicates no water supply
relationship. $r_{nz}$ ($10^4$ m$^3$) is the amount of water supplied from the $n$th reclaimed water
source to the $z$th inland tributary; $z = 1,2,...,Z$; $Z$ is the total number of inland tributaries
requiring ecological flow compensation; and $\theta_{nz}$ is the water supply relationship
between the $n$th reclaimed water source and the $z$th inland tributary.
**2.2.2 Constraints**
**Conventional water supply constraint**
According to conservation of mass, the total daily amount of conventional water
allocated in the second layer should be less than the total monthly amount of
conventional water allocated in the first layer, as described in Eq. (27):
$$\sum_{t=1}^{T}\sum_{i=1}^{I}\sum_{j=1}^{J} x_{ij}^t \alpha_{ij} + \sum_{t=1}^{T}\sum_{e=1}^{E}\sum_{j=1}^{J} x_{ej}^t \beta_{ej} \geq \sum_{t=1}^{T}\sum_{m=1}^{M}\sum_{j=1}^{J}\sum_{z=1}^{Z} q_{jz}^{t,m}\chi_{jz}, t=1,...,T \tag{27}$$

**Unconventional water constraints**
The two types of unconventional water have separate constraints. For reclaimed water
supplied to water users, the amount should satisfy the relevant water recycling standard.
The ecological water used to replenish inland tributaries is mainly pumped from





reclaimed water works and main rivers. Therefore, this replenished volume is limited
by the pumping capacity. The constraints for unconventional water are shown in Eqs.

(28)-(29):

$$\sum_{t=1}^{T}\sum_{n=1}^{N}\sum_{j=1}^{J} r_{nj}^{t}\,\theta_{nj} + \sum_{t=1}^{T}\sum_{n=1}^{N}\sum_{z=1}^{Z} r_{nz}\,\theta_{nz} = \sum_{t=1}^{T}\sum_{i=1}^{I}\sum_{j=1}^{J} x_{ij}^{t}\,\delta_{ij}\eta_{ij} + PU \qquad (28)$$


$$PU = \sum_{t=1}^{T}\sum_{m=1}^{M}\sum_{p=1}^{P} Q_{t,m}^{p,s}/10^{4} \qquad (29)$$


where $\delta_{ij}$ is the sewage discharge coefficient, which is the proportion of the water
supplied from sewage discharge; $\eta_{ij}$ is the sewage water reuse rate, which is the
proportion of reused water in the total volume of sewage water; $PU$ ($10^4$ m$^3$) is the
amount of water pumped from the main river; and $Q_{t,m}^{p,s}$ (t/d) is the flow through the
$s$th pumping station on the $m$th day at time step $t$.
**Pumping constraints**

$$Q_{t,s}^{p} \le Q_{\max,s}^{P} \qquad (30)$$


$$Q_{t}^{P} = \sum_{s=1}^{Np} r_{t,s}^{p} \qquad (31)$$


where $Q_{\max,s}^{P}$ (t/d) denotes the upper flow boundary of the $s$th pumping station; $r_{t}^{s}$ (t/d)
is the power of the $p$th pump installed at the $s$th pump station; and $Np$ is the number of
pumps stalled at the $s$th pump station.
**Water quality constraint**
To control the impacts of various point and nonpoint sources on receiving water bodies
in cities, water authorities impose water quality standards for the management of river



basins. These standards seek to maintain the water quality at a desired target level by
defining discharge limits for conventional, specific, or priority pollutants. To satisfy the
relevant standards, the following water quality constraint is established:
$$\sum_{t=1}^{T}\sum_{i=1}^{N}\sum_{j=1}^{M}\left(x_{ij}^{t}\delta_{ij}\psi_{ij}h_{j}^{u}-x_{ij}^{t}\delta_{ij}\eta_{ij}h_{j}^{u}\right)\times 10 \le H^{u} \qquad (32)$$

where $\psi_{ij}$ is the sewage water treatment rate, which is the proportion of sewage water
that is treated; $h_{j}^{u}$ (mg/L) is the concentration of the $u$th contaminant per unit treated
water required by the $j$th user; and $H^{u}$ (kg) is the upper limit of the $u$th contaminant
allowed to be discharged in the study area.

## 2.3 Third layer of the PTSOA decision-making model

After obtaining the results for the former two stages of the allocation process and the
two levels of decision-making, the third model layer is constructed to achieve regional
synergy s. It refers to the collaborative allocation of water resources in different
subregions of a city, and it is intended to balance and maximize the interests of each
subregion as much as possible. Additionally, the needs of different kinds of water users
in different subregions can be met to the greatest extent possible with this approach.
Therefore, the three dimensions of synergy are also fused in this layer. The third stage
of the process (the water in different departments is supplied to different kinds of water
users, namely, residential users, industrial users and municipal users, in different
subregions) is optimized in this layer. After department-level decision-making,



conflicts of interest inevitably occur among various water users in different subregions
of a city. Therefore, the third layer considers regional-level decision-making to to
coordinate water needs and avoid conflicts of subregions in the city. Moreover, the
various development priorities of subregions are emphasized by adjusting certain
hyperparameters in the third layer. This layer is established based on the allocation
scheme obtained in the second layer of the hierarchy, and the time scale of this layer is
the same as that of the second.

Although water pollutants are controlled in the second layer, the detailed spatial

distribution of pollutants remains unknown. If one of the subregions emits a greater
pollution load than others such that the river pollution limit is exceeded, it constrains
sustainable development and undermines the fairness of the allocation. To ensure the
coordination of water quality among regions, the representative pollutant concentration
of the main reach in each subregion after configuration should meet the relevant
environmental capacity requirements. If these requirements are not met, then the
objective function for this subregion will call for a punishment, and more
environmentally friendly plans will be searched. After sewage with pollutants is
transported from outlets to water bodies, advective transport, longitudinal dispersion
and transverse mixing will occur. At the same time, physical, chemical and biological
interactions will occur in the water body. To objectively describe the degradation of
pollutants in water, it is necessary to use mathematical models to simulate physical
dynamics. Due to the heterogeneity of pollutants entering water bodies and the





uncertainty of hydrological processes, it is usually of little practical significance to
calculate the change in river water capacity over time. A steady-state model is therefore
used to calculate the water capacity of the target water body (Cetintas et al., 2010;
Zhang et al., 2019). When water quality changes are studied at the annual scale and
complete mixing is assumed, the following equation can be used to describe the water
quality change, as shown in Eq. (33):

$$\frac{V dc}{dt} = Q(Ce - C) + Sc + r(c)V \tag{33}$$

where $V$ (m³) is the volume of water; $Q$ (m³/a) is the flow in and out of the system
at equilibrium; $Ce$ (g/m³) is the contamination concentration in the inflow (g/m³); $C$
is the pollutant concentration; $Sc$ denotes other external pollution sources (m³/a); and
$r(c)$ is the reaction rate of pollutants in water (g/m³/a). The above equation can be
defined as the basic mass balance of a water body in a completely mixed system.
Because the pollutants are evenly mixed in each small interval, the horizontal and
vertical concentration gradients of pollutants can be neglected. Therefore, the model of
water quality in mixed rivers under steady-state design conditions is adopted (Yue et
al., 2021):

$$W_c = 31.54 * \left[ C_s \cdot (Q_P + Q_E + Q_S) - Q_P \cdot C_P \right] \tag{34}$$

where $W_c$ represents the water environmental capacity (t/a); $Q_P$ is the flow in the
reach (m³/s); $C_P$ is the pollutant concentration in the river (mg/L); $Q_E$ is the sewage
discharge (m³/s); $Q_S$ is the total flow of nonpoint sources into the reach above the





control section (m³/s); and $C_s$ is the target concentration of river pollutants (mg/L).
The result calculated based on the total hydrological capacity standard is often
relatively large, which is generally referred to as nonconservative. To conform to real
conditions, the concept of a nonuniformity coefficient is introduced for correction:
$$W_c^{'} = \alpha \cdot W_c = 0.6 \cdot W_c \tag{35}$$

This coefficient is used to assign a punishment if the water quality exceeds the
relevant value in a given subregion. Based on the coefficient value, the objective
functions and constraints are adjusted accordingly. Finally, the daily decision
alternatives for water allocation from water departments to water users are obtained at
the regional decision level.

### 2.3.1  Objective function

**Regional objective function: Maximization of the comprehensive benefits of each subregion**

$$\max f_3(x) = \sum_{t=1}^{T}\sum_{i=1}^{I}\sum_{j=1}^{Jr} x_{ij}^t b_{ij} - \sum_{j=1}^{Jr}\left(D_j - \sum_{t=1}^{T}\sum_{i=1}^{I} x_{ij}^t \alpha_{ij}\right)\times \omega_j - P_r(r_{nz}) - G_r(x_{ij}^t)\times q \tag{36}$$

$$P_r(r_{nz}) = e_i \times \sum_{p=1}^{Pr} P_p^{pump}\times \nabla t_r + x_{ij}^t \delta_{ij}\psi_{ij}\omega_{ij} \tag{37}$$

$$\nabla t_r = \sum_{t=1}^{T}\sum_{n=1}^{N}\sum_{z=1}^{Zr}\left\{\left(l_{nz} + \left(r_{nz}^t \theta_{nz}/CAS_{nz}\right)\right)\Big/\left(Q_{nz}^{\max}/CAS_{nz}\right)\right\}\Big/3600 \tag{38}$$

$$G_r(x_{ij}) = \sum_{z=1}^{Zr}\sum_{u=1}^{U}\left(Q_{z,u,r}^{final} - Q_{z,u,r}^0\right) \tag{39}$$

where $b_{ij}$ (yuan/m³) is benefit per unit of water supply for the $j$th user; $\omega_j$ is the



penalty coefficient per unit of water deficiency; $j=1,2,...,Jr$; $Jr$ is the number of water
users in $r$th subregion; r= 1,2,…,R; $P_r\left(r_{nz}\right)$ is the penalty function for cost in the $r$th
subregion; $e_i$ (yuan/kW·h) is the unit electricity fee; $P_p^{pump}$ (kW·h) is the electrical
power consumed by the $p$th pump at a pump house in each hour; $p$ ranges from 1 to $Pr$;
$Pr$ is the total number of pumps in the $r$th subregion; $\nabla t_r$ (h) is the time required for
water transfer to provide support for the inland river flow in the $r$th subregion; $\omega_{ij}$
(yuan) denotes to the fee paid for sewage treatment; $l_{nz}$ (m) is the length of a water
diversion pipe from reclaimed water source $n$ to the $z$th inland river; $z$ ranges from 1 to
$Zr$; $Zr$ denotes the number of inland rivers in the $r$th subregion; $CAS_{nz}$ (m²) is the
cross-sectional area of a pipe from the $n$th reclaimed water source to the $z$th inland river;
$Q_{nz}^{max}$ (m³) is the maximum overflow capacity of the diversion pipe from the $n$th
reclaimed water source to the $z$th inland river; $G_r\left(x_{ij}\right)$ is the penalty function for
substandard water quality in the $r$th subregion; $Q_{z,u,r}^{final}$ (mg/L) is the final concentration
of the $u$th pollutant in the control section of the $z$th inland river in the $r$th subregion
after optimal configuration; $Q_{z,u,r}^{0}$ (mg/L) is the initial concentration of the $u$th
pollutant in the $z$th inland river in the $r$th subregion; and $q$ is the penalty coefficient
for substandard water quality in the $r$th subregion. The number of objective functions
in this layer depends on the number of subregions divided in the city, which is based on
local conditions.





**2.3.2 Constraints**
**Water quality constraints**
Mathematical models are often developed to help satisfy the water quality standards at
monitoring points (Zhang et al., 2019; Pourshahabi et al., 2020; Friesen et al., 2017).
However, for some cities with very few monitoring points, such approaches may lead
to good water quality in the monitored sections and poor water quality in other sections.
In these circumstances, the quality of water bodies in each subregion of a city is not
simultaneously maintained. To maintain the water quality in all subregions of a city at
the desired target level, the water quality constraint in Eq. (40) is established:
$$Q_{z,u,r}^{final} \leq Q_{z,u,r}^{control} \qquad\qquad (40)$$
where $Q_{z,u,r}^{control}$ (mg/L) denotes the control standard for the $u$th pollutant in the control
section of the $z$th inland river in the $r$th subregion.
**2.4 Model solution**
**2.4.1 Synergy degree evaluation**
Enhancing the understanding of the synergy among water allocation alternatives to
achieve broad coordination and equilibrium is crucial. The evaluation of the synergy of
a water system is strongly related to multiple complex interactions, such as the
interactions among different processes, users, and regions. However, these interactions
have rarely been explicitly captured in prior evaluations of water allocation. One of the


588 key network metrics used in network analysis, connectivity, is a promising measure of

589 the degree of coordination among different objectives in complex systems (Weitz et al.,

590 2018). Connectivity reflects the connectedness of a given link to all possible links in

591 the network, and the strength of each link is weighted, reflecting the number and

592 strength of correlations (Felipe-Lucia et al., 2020). In this study, connectivity is used to

593 embody coordination in the context of synergy, as shown in Eq. (26). Due to the limited

594 supply of water resources, competition among different objectives is unavoidable, and

595 the objectives cannot be fully optimized to equal extents, i.e., , an increase in one target

596 output may decrease another output. Therefore, equilibrium is integrated as another

597 vital part of the synergy devoted to maintaining a balance among the satisfaction of

598 each goal in a system. The equilibrium based on the principle of information entropy

599 (Gao et al., 2013; Zivieri, 2022) is shown in Eq. (27). Information entropy is a measure

600 of the uncertainty associated with a random variable and is used to quantify the

601 information contained in a message, usually in bits or bits/symbols; furthermore, it has

602 been widely used to represent the fairness or equilibrium of a system (Chen et al., 2022;

603 Zhao et al., 2022). When $H$ is low, the level of equilibrium in the system is high. By

604 combining the quantification of coordination and equilibrium, the synergy degree is

605 appropriately determined (Eq. (29)). Notably, the total synergy index (*TSI*) of a system

606 is used for both generating candidate management alternatives in the generation phases

607 of PTSOA and performing assessments of the associated level of synergy, as shown in

608 Eqs. (41-44).

$$SSI_{ob_i} = \frac{\sum_{j=1}^{N} c_{ij} \times (ob_i + ob_j)}{\sum_{j=1}^{N} (ob_i + ob_j)}, i \neq j \qquad (41)$$

$$H(S) = -\sum_{i=1}^{N} \frac{(1 - u_{ob_i})}{N} \log \frac{(1 - u_{ob_i})}{N} \qquad (42)$$

$$u_{ob_i} = \frac{ob_i - ob_{i,\min}}{ob_{i,\max} - ob_{i,\min}} \qquad (43)$$

$$TSI = \frac{\sum_{i=1}^{N} SSI_{ob_i}}{H(S)} \qquad (44)$$

where $SSI_{ob_i}$ is the connectivity of the $i$th object; $c_{ij}$ is the Pearson correlation
between the $i$th object and $j$th object; $ob_i$ and $ob_j$ are the values of the $i$th and $j$th
objective functions, respectively; $TSI$ is the synergy index of the system; $H(S)$ is
the overall equilibrium of all objects based on the principle of information entropy;
$u_{ob_i}$ is the standardized value of the $i$th object; $N$ is the total number of objects in the
system; $ob_{i,\min}$ and $ob_{i,\max}$ are the minimum and maximum critical thresholds of the
parameter $ob_i$, respectively.
**2.4.2 Hierarchical optimal algorithm design for the PTSAO model**
Based on the algorithm design with a hierarchical objective function proposed by Li et
al. (2022), a new level is added to the original two levels of the algorithm, and the
alternative generation phase is improved for better synergy. In this algorithm, the
objective functions in the upper decision level is first satisfied, and then the lower-level
objective function provides an optimal result based on the results of optimal allocation



in the upper level. To provide as comprehensive solutions as possible, the decision
alternatives need to be classified into different sets for further selection. In addition, the
synergy degree of the result of each layer is calculated to select optimal decisions
among all Pareto front solutions. The detailed steps of the hierarchical optimal
algorithm are as follows:
I.     In the first level, calculate the objective function (city level) values for the social,

economic and sustainable development components, and sort the results with

NSGA-III (Pourshahabi et al., 2020; Chen et al., 2017) to obtain each Pareto

front $F_1$, $F_2$, …, $F_i$.

II.     Classify the Pareto fronts into $K$ ($K$ is determined based on the diversity of

policies) elements with the K-means algorithm (Liu et al., 2022), which is used

to partition a data set into $K$ distinct and nonoverlapping clusters. To perform

K-means clustering, we first specify the desired number of clusters K. Then, the

K-means algorithm is used to assign each observation to exactly one of the $K$

clusters.

III.     Calculate the synergy degree of each individual in the front, and select the

solution that yields the greatest synergy in each cluster. $K$ solutions are obtained

in the first layer.

IV.     Use the selected $K$ solutions in the first layer to establish constraints in the

second layer. Solve the objective function of the second layer with NSGA-Ⅲ.

V.     Calculate the synergy degree of each individual in the front and select the



solution that yields the greatest synergy as well as the two solutions that
maximize the conventional and unconventional water supply department
objective functions in all Pareto fronts with K preconditions.
VI.    The three selected solutions in the second layer are used to establish constraints
in the third layer. Solve the objective function of the third layer with NSGA-Ⅲ
under the three preconditions.
VII.    The synergy degree of each individual in the front is calculated, and the solution
that yields the greatest synergy in the third layer is selected. Three solutions are
obtained considering the synergy in the former two layers. Finally, the
synergistic configurations optimal for all stages in the whole process are
identified considering the synergy among decision levels, processes and time
scales.

## 659 3. Application

## 660 3.1 Study area

Yiwu city is selected as a case study to validate the applicability of the PTSOA model.
Yiwu city is in Southeast China, located from 119°49 'E-120 °17' E and 29°02 '13 "N-
29 °33' 40" N. The city covers an area of 1105 km$^2$. The area is characterized by a
scarcity of water resources, and the conventional water supply is under severe stress.
The regional water consumption depends heavily on transported water and external





water transfer. The per capita water resources total 622 m$^3$, only 22.6% of the provincial
average and 19.1% of the national average. Moreover, the problem of water pollution
has become a bottleneck constraint for the development of Yiwu city. Therefore, it
represents a typical water-scarce city with limited conventional water. Notably, water
quality in Yiwu has been subjected to significant environmental stress because of the
negative effects of wastewater discharge with the rapid development of industry. The
current water quality is poor, with Class $V$ water, and the main pollutant concentrations
exceed the corresponding standards (Zhejiang Natural Resources and Statistical
Yearbook on Environment, 2020). As shown in Fig. 3, the Yiwu River crosses the city
from northeast to southeast. Additionally, there are six ecological water compensation
outlets in six main tributaries in the Yiwu River.

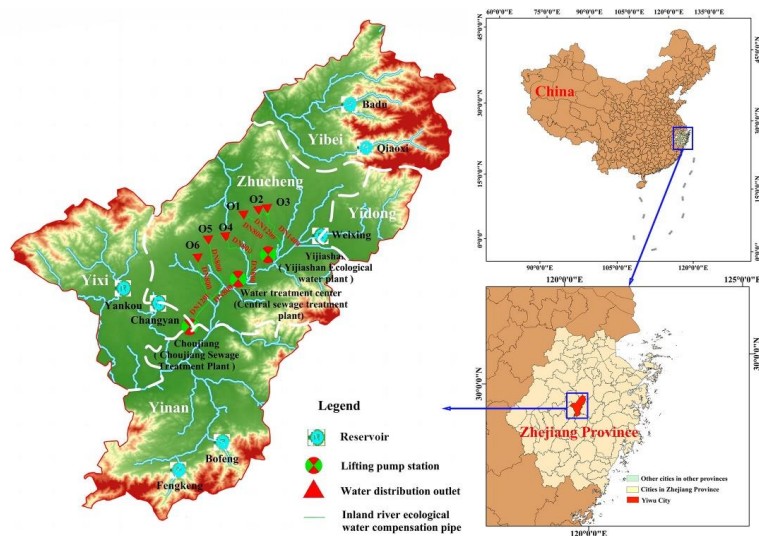

**Fig. 3. Map of the study area**





## 3.2 Generalization of the water system


An initial multisource, complementary and mutually regulated system has been
developed for Yiwu, and this system spans the entire urban water cycle (water source-
water supply-water use-drainage-drainage collection-recycling and reuse). To apply the
optimal water allocation model to the complex real-world water system, all
stakeholders in the water system should be schematized into a topological system, as
shown in Fig. 4. The diagram comprises five modules: water sources, water supply,
water use, water drainage and external discharge for all stakeholders.
The first module includes seven main reservoirs, two water diversion projects, the
Central Sewage Treatment Plant and the Yiwu River. The seven reservoirs and two
water diversion projects (as shown in Table 1) supply high-quality water. There are
complex connections between the first and second modules. For example, two
reservoirs supply water to one waterworks or one reservoir feeds two or three
waterworks simultaneously. The reservoirs also supply some of the agricultural and
ecological waters to subareas of the city. The Yiwu River, with a total length of 38.39
km and 21 first-class tributaries in the city, and the Central Sewage Treatment Plant, as
shown in Table 2, are low-quality water sources. Additionally, excluding water from
reservoirs, most agricultural irrigation water is supplied from surface water stored in
hundreds of small reservoirs and mountain ponds. Since there are no data available for
agricultural irrigation water, which accounts for only a small portion of the total water



demand in the area, this water volume is ignored in the model. For the second module,
high-quality water piped from reservoirs is transported to nine urban and rural
centralized waterworks (as shown in Table 2). The Yiwu River distributes low-quality
water to the Yijishan Ecological Water Plant and Sufu Industrial Water Plant through
the Yijishan and Baisha Water Pump Stations, respectively. The water discharged at the
Central Sewage Treatment Plant is transferred to the Choujiang Industrial Water Plant.
Based on the water supply project distribution and the economic as well as social
development levels , Yiwu is divided into five districts, as shown in Table 3: the Central
District, Yidong District, Yibei District, Yinan District and Yixi District. The third
module comprises both high-quality water users (high-quality water users consist of
urban and rural domestic water users and industrial water users in the water supply
network of urban and rural public water plants) and low-quality water users (low-
quality water users include industrial water users, municipal water users and ecological
water replenishment for inland rivers) in each district. There are nine sewage treatment
plants in the fourth module (which focuses on the drainage stage), as shown in Table 2.
The unreused water from sewage treatment plants is discharged to the external
environment. Reuse processes are also considered in the system.





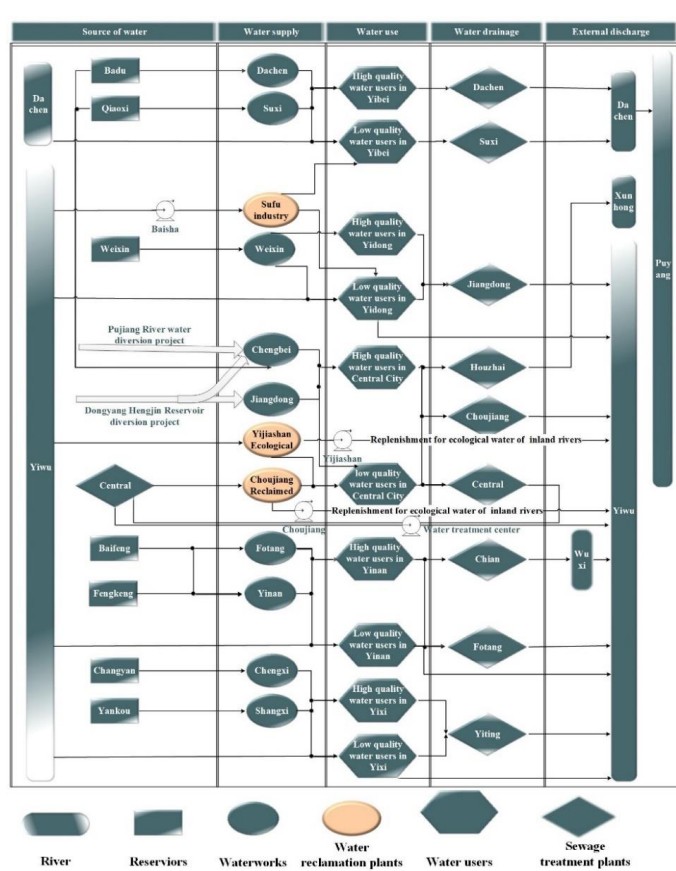


**Fig. 4. Schematic diagram of Yiwu city**

## 3.3 Parameter determination

According to the flow duration curve of the annual natural inflow data for 51 years
(1963-2014), three years with exceedance probabilities of 50%, 75% and 90% are
selected to represent normal (1984.1–1985.1, annual mean inflow: $1.33 \times 10^8$ m$^3$), dry
(2008.1–2009.1, annual mean inflow: $1.11 \times 10^8$ m$^3$), and extremely dry (1971.1–
1972.1, annual mean inflow: $0.63 \times 10^8$ m$^3$) scenarios, respectively. In addition to





inflow, the data used in the PTSOA model mainly include the data for the parameters
in each layer. Water demand values were calculated using the Yiwu City Water
Resources Comprehensive Plan 2020, as shown in Table 1.
**Table 1** Water demands of various regions in Yiwu in 2020 ($10^4$ m$^3$)

| Subregion | Yibei | Yidong | Zhucheng | Yixi | Yinan |
|---|---|---|---|---|---|
| water demand | 1695 | 572 | 11813 | 2198 | 2045 |


The water resources fees paid to the government total 0.3 yuan/m³. The parameters
of the reservoirs and external water division projects in Yiwu city are listed in Table 2.
**Table 2** Parameters of the reservoirs and external water division projects

| Reservoirs & External sources | Water Fee yuan/m³ | Initial storage $10^4$ m$^3$ | Dead storage $10^4$ m$^3$ | Flood limit storage capacity $10^4$ m$^3$ | Absolute storage capacity $10^4$ m$^3$ |
|---|---|---|---|---|---|
| Badu | 0.99 | 1359 | 49 | 2688 | 2639 |
| Qiaoxi | 1.30 | 1505 | 77 | 2933 | 2856 |
| Weixin | 0.37 | 500 | 17 | 483 | 466 |
| Baifeng | 1.05 | 1013 | 15 | 2010 | 1995 |
| Fengkeng | 1.15 | 778 | 55 | 1501 | 1446 |
| Yankou | 1.49 | 1820 | 499 | 3140 | 2641 |
| Changyan | 0.70 | 491 | 41 | 940 | 899 |
| Pujiang Project | 1.00 | 0 | 0 | 3000 | 3000 |
| Dongyang Project | 1.00 | 0 | 0 | 5000 | 5000 |


The Tennant method is applied to calculate the ecological water demand. In this
method, the relationship between the annual average discharge and habitat quality is
considered, and the percentage of the annual average natural runoff is used as the
recommended value of the ecological water demand for a given river channel.

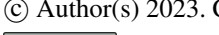



According to the recommended values, the percentage of runoff required for the fish
spawning period from April to September is 30% and the percentage runoff in the
general water consumption period (October to March) is 10%.
Based on observations obtained with the F601 evaporator (a standard evaporation
instrument widely used in China), evaporation is calculated as:
$$EP = E \times k \qquad (45)$$
where $EP$ (mm) is the evaporation of a reservoir; $E$ (mm) is the observed evaporation;
and $k$ is a reduction coefficient. According to observations, this coefficient is the same
for every reservoir and varies throughout the year (Zhao, 2014). The prices of
conventional water and reclaimed water are 1.7 and 2.6 yuan/m³, respectively.

The monthly mean monitoring data for effluent pollutant concentrations and the

daily maximum processing capacities of sewage treatment plants were obtained from
the monitoring systems of the sewage treatment plants. For example, the concentrations
of COD, NH3-N, TN, and TP in the sewage of the Jiangdong Sewage Treatment Plant
are 13.80 (mg/L), 0.22 (mg/L), 6.02 (mg/L), and 0.13 (mg/L), respectively. The daily
maximum processing capacity of Jiangdong Sewage Treatment Work is 12 ($10^4$ t/d).
The effluent quality of sewage treatment works satisfies the Class A Standard used in
China. The maximum capacities of the Baisha pump station, Yijiashan pump station,
Choujiang pump station and water treatment centre pump station are 13 t/d, 13.5 t/d, 10
t/d, and 4.5 t/d, respectively.

Additionally, the environmental capacities of the six tributaries that are replenished





with ecological water are calculated according to Eqs. (33)-(35), and the results are
listed in Table 3. COD, TP and TN are selected as representative pollutants in the
tributaries to guarantee the water environmental quality of inland rivers. The water
quality goals for the tributaries must conform to the Class III standard according to GB
5749-2006 in China. The unit electricity price of pump stations in Zhejiang Province is
0.41 yuan/kW·h. GB50014-2006 (2014 edition) stipulates that the comprehensive urban
domestic sewage quota should be 80-~90%, and the urban comprehensive domestic
sewage quota should be 90% in areas with extensive drainage facilities. According to
the "Yiwu Water Resources Bulletin 2020", the urban comprehensive domestic sewage
quota is set to 90%, and the sewage treatment rate is set to 100%. The benefits per unit
water supply for different users in different subregions are determined from the Yiwu
Water Price Adjustment Plan 2020.

**Table 3** Area and environmental capacity of tributaries

| Name of tributary | Area (km²) | Class III | | |
|---|---|---|---|---|
| | | COD (t/a) | TN (t/a) | TP (t/a) |
| Chengdong | 3.4 | 188.1 | 4.7 | 0.4 |
| Chengzhong | 8.7 | 432.7 | 31.5 | 3.8 |
| Chengxi | 6.3 | 302.5 | 9.5 | 2.3 |
| Chenganan | 7.1 | 318.8 | 0 | 3.6 |
| Hongxi | 12.5 | 778.8 | 138.8 | 7.9 |
| Dongqingxi | 38 | 1271.4 | 221.5 | 12.7 |

## 4. Results and discussion

By solving the PTSOA model for Yiwu city, synergistic optimal water allocation results
for different layers (across different decision levels, water use sectors, and subregions)



are obtained under normal, dry and extremely dry conditions. Pareto sets are obtained
across 500 runs of the PTSOA model with the proposed hierarchical optimization
algorithm.

## 4.1 Results of the first layer of the PTSOA model for synergistic optimal water allocation

To demonstrate the relationship among conflicting objectives, sets of Pareto solutions
for the first layer under normal, dry and extremely dry conditions are shown in Fig. 5.
The optimization using the Pareto concept allows the operator to choose an appropriate
solution depending on the prevailing circumstances and analyse the trade-off among
the conflicting objectives. In each of the figures, the total water supply shortage, total
water supply benefit and total amount of water retained in reservoirs in Yiwu city are
plotted. The colour of the markers indicates the classification of the solutions of the K-
means method, as described in Sect. 2.4.2. All of the decision alternatives are classified
into six groups marked in different colours for broad-scale decision-making. The names
of the classes are marked in the figure in red (for example, K1-1 represents the first
class of solutions in the normal scenario, and K3-2 represents the second class of
solutions in the extremely dry scenario). The red arrows indicate optimization
directions. The ideal solution is located at the top-right corner (low total water supply
shortage, high total water supply benefit, and relatively high total amount of reserved
water in reservoirs) of the plot. The geometries of the tradeoffs vary significantly across





the applications, as is expected given the different hydrological conditions. Generally,
the total water supply shortage and the total amount of water retained in reservoirs show
an inverse relationship. In contrast, the total water supply benefit shows a direct and
positive influence on the total water supply shortage. The water supply reliability of the
selected decision alternatives is greater than 95% under normal, dry and extremely dry
conditions. The total amount of reserved water in reservoirs under normal scenarios
varies in the range of $2.91\times10^7$ m³ to $6.14\times10^7$ m³, which is much higher than that under
the extremely dry scenario, with a value of $1.44\times10^7$ m³ to $2.93\times10^7$ m³. This finding
demonstrates that the optimal allocation is able to reconcile the present demand and
future needs, even in extremely dry scenarios. The total water supply shortage in all
scenarios is less than 5% of the water demand, which indicates that the guaranteed water
supply is greater than 95%.

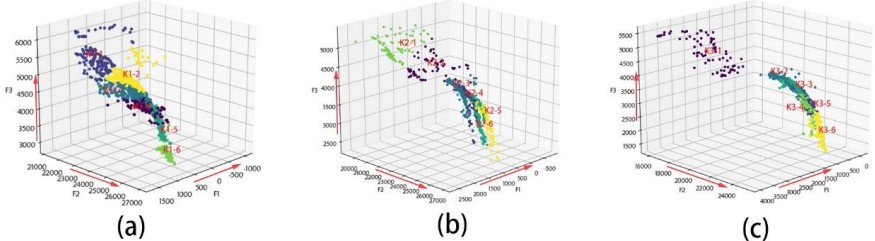

(a)       (b)       (c)


**Fig. 5.** Sets of Pareto solutions after 500 model simulations with the hierarchical
optimal algorithm under (a) normal, (b) dry and (c) extremely dry scenarios. The red

arrow indicates the direction of optimization.


We further present the TSI (total synergy index), SSI (total connectivity) and H



(overall equilibrium) values for different classes characterized based on the optimal
PTSOA solutions under three scenarios, as shown in Fig. 6. In the PTSOA model, the
Pareto solutions with the best TSI values are input to the second layer for further
optimization. Thus, the red points in Fig. 6 represent the selected schemes for all classes.
We observe that the variation in the TSI is consistent with that in the SSI in some, but
not all cases. In some cases, difference are mainly caused by the influence of H, which
influences the optimal hydrological equilibrium, especially in dry conditions. Although
normal conditions are most conducive to achieving equilibrium, the better H value in
extremely dry conditions than in dry conditions seems nonintuitive. However, these
results suggest that when water is very limited, equally limited water is supplied to all
users, thus enhancing the overall equilibrium. We note that the SSI is higher in the
normal scenario than in the other two scenarios. We attribute this to relatively abundant
water being useful for stakeholders to achieve synergy due to the reduced competition
compared to other cases. The TSI values reach maximums of 5.36, 7.37 and 10.82 under
normal, dry and extremely dry conditions, respectively. Since the TSI is used to
illustrate the synergy of allocation plans under certain conditions, the three kinds of TSI
values are not comparable.



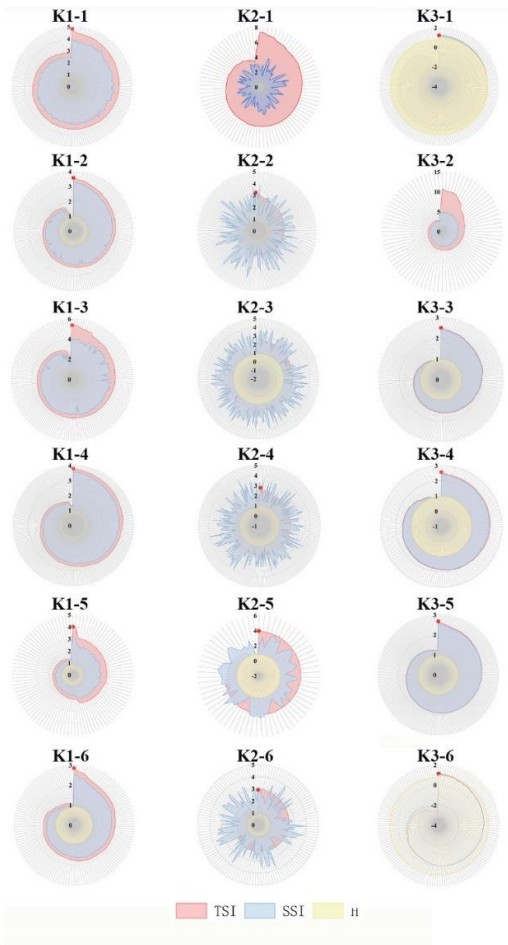


**Fig. 6.** Comparison of TSI (total synergy index), SSI (total connectivity) and H

(overall equilibrium) values among various Pareto solutions in different classes for

the (K1) normal, (K2) dry, and (K3) extremely dry scenarios.

As an example, Fig. 7 provides the specific water supply decision alternatives for
the first layer that maximize synergy in each cluster under normal conditions. The water
allocation plans for the seven main reservoirs and two external water diversion projects
in every month of the configuration period are displayed. All reservoirs and water works





are represented by abbreviations based on their full names in Fig. 7. For example, QX-
CB is the label for the water supplied from Qiaoxi Reservoir to Chengbei Water Works.
The water volumes supplied by Qiaoxi Reservoir to Chengbei Water Works (ranging
from $1.78×10^7$ m³ to $3950×10^4$ m³) and from the Pujiang External Water Division
Project to Chengbei Water Works (ranging from $2.57×10^7$ m³ to $3×10^7$ m³) are relatively
high in all clusters. This result is consistent with the fact that Chengbei Water Works is
one of the main conventional water sources for the central city area, a region that
accounts for more than 50% of the total water demand of Yiwu city. The water supplied
by the two external water diversion projects from August to December is higher than
that in other months. The mean monthly precipitation in these months is only 58-74%
of the mean annual precipitation in Yiwu, so more external water is supplied for
replenishment. Baifeng and Fengkeng Reservoirs supply similar volumes of water to
their two connected waterworks.





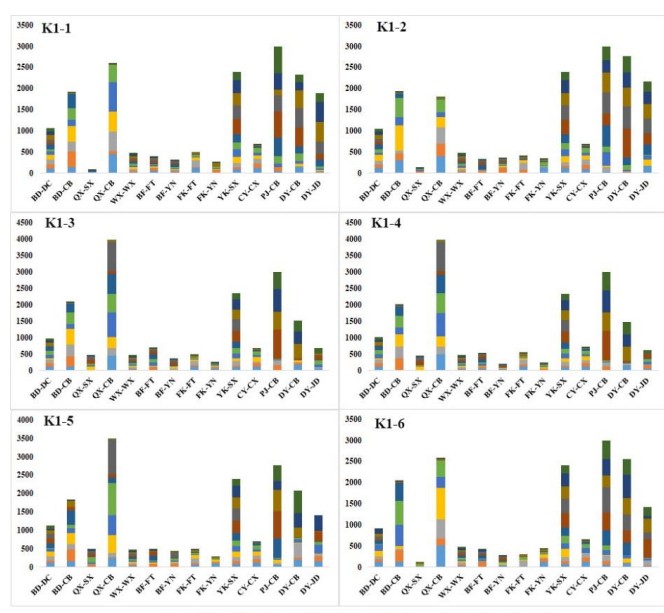

**Fig. 7.** Water supply from each reservoir to connected water works in each month in

the normal scenario ($10^4\,\mathrm{m}^3$)

## 4.2 Results of the second layer of the PTSOA model for synergistic optimal water allocation

The 6×3 decision alternatives selected in the six clusters of the optimal first-layer results

in the normal, dry and extremely dry scenarios are input into the second layer for further

optimization. As shown in Fig. 8, the total amount of water retained in water works and

the amount of unconventional water supplied show a negative correlation. In the

alternative generation phase of game bargaining between the two objectives, the greater

the total amount of water retained in water works is, the greater the amount of

unconventional water supplied will be, which indicates that more conventional water



will be saved when more unconventional water is supplied. Conversely, the amount of
unconventional water supplied is affected by the total amount of water retained in water
works.
In the second layer, three alternatives in each scenario are selected as prior
conditions for further optimization. In addition to the two individual extrema of the two
objectives, the alternative that yields the best synergy is also identified, and it is similar
to that in the first layer. In the normal scenario, the *TSI* values are -0.90, -1.02 and -0.88
in the cases with the optimal conventional water supply, unconventional water supply
and synergy, respectively. The most synergistic approach includes only $7.08{\times}10^4$ m³
more conventional water retained than that in the conventional water supply cases and
only $9.72{\times}10^4$ m³ more than that in the optimal unconventional water supply case.
Therefore, not only is the best *TSI* value obtained, but the requirements of both
conventional and unconventional water supply departments are met. The TSI of the
most synergistic solution is the highest under dry conditions, with a value of -0.79.
Overall, the total amount of water retained in the water works ranges from $3.95{\times}10^7$
m³ to $5.75{\times}10^7$ m³, $3.12{\times}10^7$ m³ to $5.31{\times}10^7$ m³, and $2.43{\times}10^7$ m³ to $3.96{\times}10^7$ m³ for the
three types of conditions. The total amount of unconventional water supplied ranges
from $5.95{\times}10^7$ m³ to $7.48{\times}10^7$ m³, $6.34{\times}10^7$ m³ to $7.56{\times}10^7$ m³, and $6.28{\times}10^7$ m³ to
$7.37{\times}10^7$ m³ in the normal, dry and extremely dry scenarios, respectively. It is notable
that the drier the conditions are, the lower the amount of water retained in water works
and the greater the amount of unconventional water supplied. This approach is useful





for cities to mitigate the risk of drought. Additionally, based on the constraints regarding
the contaminants allowed to be discharged, more than 1272.21 t and 48.81 t of COD
and ammonia nitrogen emissions are avoided per year. In other words, the balancing of
the two objectives is beneficial for managers to determine an equilibrium solution that
satisfies the relevant demand and successfully avoids surplus conventional or
unconventional water supply in terms of sustainable development.

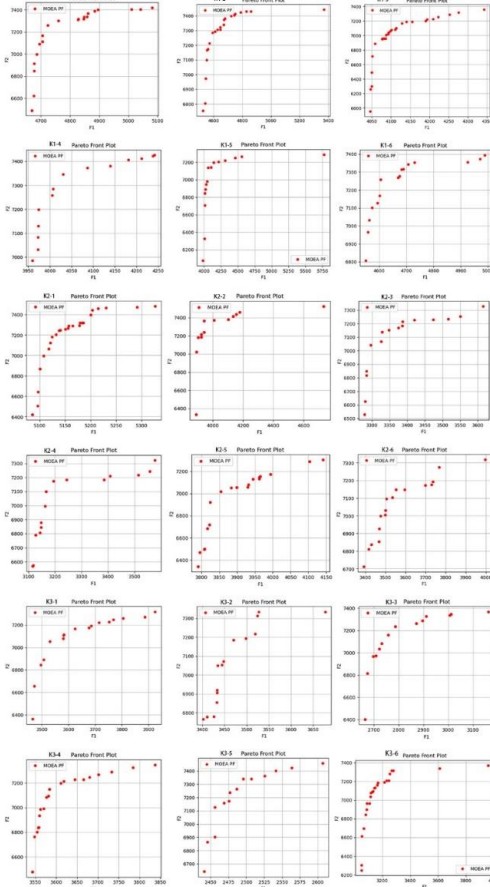


**Fig. 8.** Pareto fronts of the second layer in the PTSOA model after 500 simulations

with the hierarchical optimal algorithm in the normal, dry and extremely dry

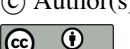



scenarios. F1 represents the total amount of water retained in water works ($10^4$ m$^3$),
and F2 represents the amount of unconventional water supplied ($10^4$ m$^3$). The
direction of optimization is from the top-right corner to the bottom-left corner.

## 4.3 Results of the third layer of the PTSOA model for synergistic optimal water allocation

**4.3 Results of the third layer of the PTSOA model for synergistic**
**optimal water allocation**
After selecting the three scenarios that yield the best synergy and the two best objective
functions for characterizing all Pareto fronts of the second layer in each scenario, these
3×3 solutions are input to the third layer for further optimization. Fig. 9 shows the
tradeoffs among the five objectives in the third layer of the PTSOA model for the (S1)
normal, (S2) dry, and (S3) extremely dry scenarios (these abbreviations are used to
distinguish these results from those of the above two layers). The number following the
'-' represents the selected solution from the second layer. For example, S1-1 represents
the normal scenario with the minimum total amount of water retained in water works,
S1-2 represents the normal scenario with the maximum unconventional water supply
and S1-3 represents the normal scenario with the maximum synergy degree in the
second layer. In each of these plots, the abscissa denotes the identifier for the objective
functions, which ranges from 1 to 5, and the ordinate gives the objective values in the
Pareto fronts ($10^4$ yuan). The five dimensions include the comprehensive benefits of
the Yibei (1.0 dimension), Yidong (2.0 dimension), Yixi (3.0 dimension), Yinan (4.0
dimension) and central city (5.0 dimension) subregions. As shown in the figure, the



central city achieves the most comprehensive benefit among the five cities. This is
primarily attributed to the large population and intensive industry in this area. However,
the benefits in the other four subregions are also high compared to recent levels and
those achieved with traditional allocation methods, as shown in Table 9. Interestingly,
the comprehensive benefits in the subregions are greater in the scenario with the
maximum synergy degree under normal conditions than in the other two scenarios.
Technically, the total comprehensive benefits in the five subregions in this scenario are
approximately $2.3 \times 10^8$-$5.1 \times 10^8$ yuan higher than those in other cases, which indicates
that the solution with the highest synergy degree in the second layer is the best choice
for managers in normal years. However, the various subregions obtain the greatest
benefits when maximizing the unconventional water supply in dry and extreme
scenarios. This result indicates that increasing the use of unconventional water in dry
and extremely dry years would significantly increase the potential benefits.

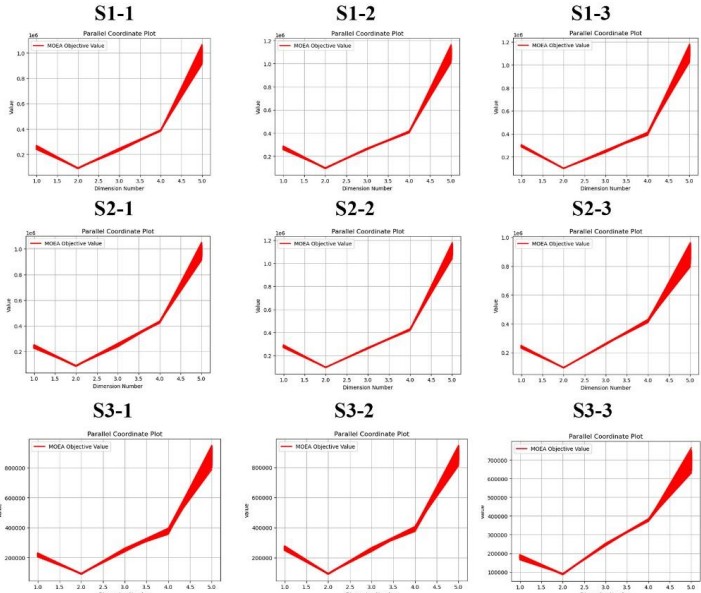

**Fig. 9.** Illustration of parallel-reference Pareto sets from the third layer in the

PTSPOA model attained across all runs for the (S1) normal, (S2) dry, and (S3)

extremely dry scenarios

Fig. 10 presents the optimal comprehensive benefit in each subregion. In all

scenarios, the central city is associated with the highest comprehensive benefit,

followed by Yixi and Yinan, and the comprehensive benefit in Yidong is relatively low.

This result may be related to this subregion having the smallest area (72.2 km$^2$) and the

smallest population (7.7×10$^4$ people). The comprehensive benefits vary among different

solutions and scenarios. Among the three normal decision alternatives, F1, F2 and F5

are highest in S1-3, with values of 3.03×10$^9$ yuan, 9.90×10$^8$ yuan and 1.12×10$^{10}$ yuan,

respectively. This indicates that considering the synergy degree could increase the



comprehensive benefit in most subregions in the normal scenario. Among the
alternatives in the dry and extremely dry scenarios (excluding F4 and F5), other
objectives are highest in S2-2, with values of $2.84\times10^9$ yuan, $9.63\times10^8$ yuan and
$2.67\times10^8$ yuan, respectively. It suggests that maximizing the unconventional water
supply is beneficial for the system in dry conditions. Additionally, F4 is highest, with a
value of $2.29\times10^9$ yuan, in S2-3 among the three solutions in the dry scenario, and F5
is highest, with a value of $9.17\times10^9$ yuan, in S3-1 in the extremely dry scenario.

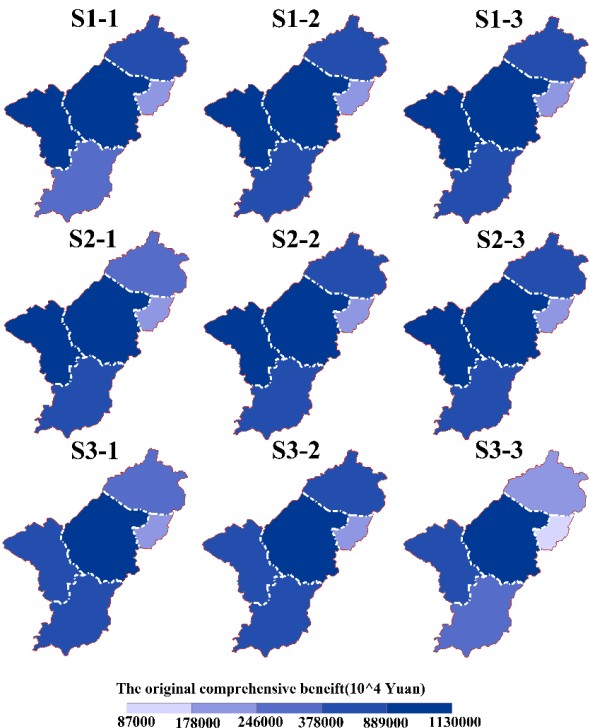


**F10.** Comprehensive benefit in each area after the regional collaborative allocation of
water resources



## 4.4 Discussion


To assist policymakers in understanding the complex and systemic nature of water
systems and reveal the dynamic interactions among objectives, network analysis and
optimization was applied. By revealing the interactions among different objectives, we
determine the level of synergy in complicated water systems, identify the challenges
and opportunities for sustainable development of water systems in cities with various
subregions, and provide valuable insights and specific action priorities for these regions.
In the networks shown in Fig. 11, each node represents an individual objective (F1, F2,
F3, F4, and F5 represent the comprehensive benefits in Yibei, Yidong, Yixi, Yinan and
the central city, respectively), and pairwise objectives that are significantly ($P < 0.05$)
correlated are connected by a link, where the strength of each link is related to the
Pearson correlation coefficient. The obtained networks with 5 nodes were weighted and
undirected (directionality can be estimated only if the direction of causality is known).
The size of the circles in the figure indicates the connectivity of each objective. We
considered trade-offs (i.e., negative correlations wherein one objective improves while
the other worsens) among the objectives. In most scenarios, F5 was the relatively
dominant objective, signifying that other objectives disproportionately deteriorated as
progress was made towards the benefit of the central city, as shown in Fig. 11. It is
evident that the trade-offs are more balanced in the scenarios with the highest degrees
of synergy (S1-3, S2-3, and S3-3), which indicates that the tradeoffs and competitions



among the objectives are alleviated when synergy is considered. The links show that
the conflicts of interest between F4 and F5 in scenarios S1-1 and S2-2 are extremely
notable, suggesting that the comprehensive benefits in Yinan and the central city
correspond to strong negative interactions in these cases. The connectivity of most
objectives was relatively low in the tradeoff network in the extremely dry scenario, but
F5 played a dominant role in terms of negative interactions among objectives, although
the connectivity of F5 was lower than other connectivities in most normal and dry
scenarios. Moreover, as the scenario varied from normal to extremely dry, the impact
of individual regional targets on the whole system diminished.

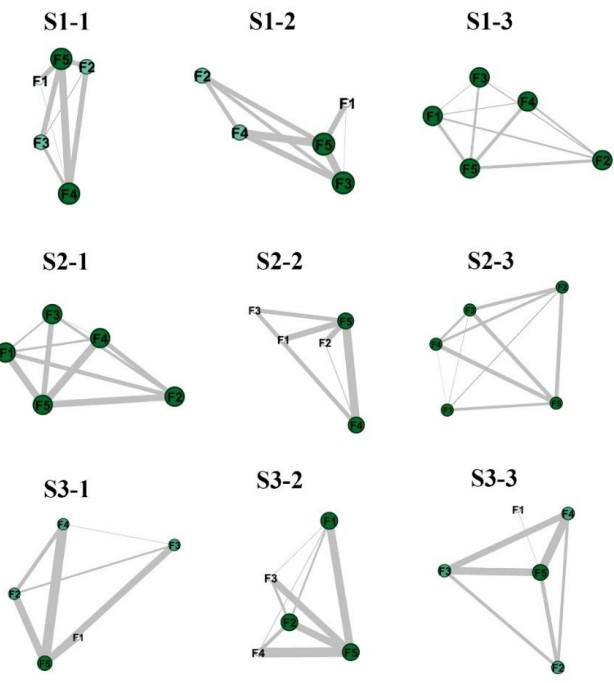


**Fig. 11.** Network analysis of the results of layer 3






For comparison, we applied five widely used MOEAs, namely, NSGA-II, SPEA-

II, ε-MOEA, IBEA, and MOEA/D, to solve cases with 3+2+5 mathematical objectives
(3 objectives in the first layer, 2 objectives in the second layer and 5 objectives in the
third layer) with the same constraints given previously for Yiwu city under normal, dry
and extremely dry conditions. The constraints and common parameters, such as the
maximum number of model simulations and the simulated binary crossover (SBX) rate,
are set to those used in the PTSOA model. However, it is difficult to determine feasible
decision alternatives with MOEAs, even though the number of iterations is increased
to 20000 (which is far beyond that considered in the previous modelling) because the
complexity of the system overshadows the optimization capabilities of these traditional
models. These results reconfirm the superiority, efficiency and decoupling capability of
the proposed model for optimal allocation cases involving complex water systems with
multiple stakeholders, multiple sources, multiple decision-makers and embodied reused
systems. By embedding the targets into hierarchical layers, the excessive abandonment
of some promising alternatives is avoided, and optimal allocation is progressively
achieved. In general, the hierarchical structure of the PTSOA model can simulate
complicated systems with multiple complex objectives and constraints.

In addition, the five MOEAs were used to solve the equations in the third layer of

the PTSOA model, and the overall targets in the first layer were determined based on
these solutions. The necessary parameters and hyperparameters were consistent with





those used in the third layer of the PTSOA model. Additionally, the benefits in the
current case with no optimization calculated based on the actual water supply are given
for comparison. The current situation was categorized as a normal scenario, and other
models were established with the same conditions to facilitate further comparison and
analysis. There were distinct decision alternatives generated by each model, and the
relevant results are listed based on their value ranges. As shown in Table 4, although
NSGA-II and ε-MOEA yield slightly higher F2 values than PTSOA and F3 generated
by IBEA ($4.8 \times 10^8$ -$7.2 \times 10^8$ yuan) is higher than obtained with PTSOA, PTSOA
performs better than other models in most cases. The PTSOA is shown to be the best
model for obtaining comprehensive benefits for the subregions in Yiwu in the normal
scenario, demonstrating that the PTSOA model offers advantages including identifying
the best alternatives and achieving greater subregional benefits than the other models.
The proposed model yields a $1.76 \times 10^9$-$15.67 \times 10^9$ yuan total comprehensive benefit
improvement and can save approximately $3.2 \times 10^7$- $4.7 \times 10^7$ (m³) of conventional water
compared to the current values. It is also evident that the proposed model yields the
highest TSI values, reflecting the improvement achieved by considering the synergy of
the system. In terms of the targets in the first layer, except MOEA/D, other traditional
models fail to retain enough water (water requirements for living under extreme drought
conditions of the next configuration period) in the reservoirs to meet future basic needs.
For MOEA/D, although it generates a slightly higher total water supply benefit, with a
value of $2.81 \times 10^8$-$3.12 \times 10^8$, the total water supply shortage and the total amount of





reserved water in the reservoirs are worse than the amounts obtained with the proposed
model. PTSOA trades some economic benefits for enhanced water supply reliability
and sustainable development, resulting in a decrease in the water supply from
conventional water plants.
However, the consideration of reclaimed water in the proposed model effectively
reduces the use of traditional water and improves the quality of the water environment
by reducing sewage discharge, and other benefits are also achieved (such as meeting
the quality standards for river water and guaranteeing that the ecological water demand
of inland rivers is met). The results obtained by the PTSOA may help guide both the
government and general public. Our proposed model is superior to traditional models.
It can not only optimize water resource utilization and secure water supplies but also
enhance the synergy and environmental quality of water systems. Considering synergy
across various time scales, the proposed model ensures the synergistic allocation of
water resources at yearly, monthly and daily scales while securing both present and
future water supplies.
**Table 4** Comparison of the comprehensive benefits in the five regions (F1, F2, F3, F4,
and F5) and the TSI values in the current situation and obtained using NSGA-II,
SPEA-II, ε-MOEA, IBEA, MOEA/D, and PTSOA in the normal scenario

| Comparison | Comprehensive benefits ($10^9$ yuan) | | | | | TSI |
|---|---|---|---|---|---|---|
| | F1 | F2 | F3 | F4 | F5 | |
| NSGA-II | 2.72~2.86 | 0.91~1.03 | 2.57~2.60 | 3.21~3.37 | 7.38~9.95 | -3.13~-2.82 |
| SPEA-II | 2.84~2.97 | 0.93~0.99 | 2.58~3.15 | 3.02~3.68 | 8.22~9.99 | -2.39~-2.46 |





| | | | | | | |
|---|---|---|---|---|---|---|
| ε-MOEA | 2.47~2.33 | 0.85~1.12 | 2.21~2.32 | 3.05~3.18 | 9.23~9.91 | -3.41~-3.06 |
| IBEA | 2.57~2.88 | 0.87~0.92 | 3.05~3.11 | 3.20~3.32 | 5.27~8.28 | -3.28~-3.11 |
| MOEA/D | 2.55~2.90 | 0.99~1.02 | 3.15~3.20 | 3.34~3.36 | 9.82~10.11 | -2.37~-1.54 |
| Current situation | 2.05 | 0.83 | 2.49 | 3.11 | 9.87 | -3.20 |
| **PTSOA** | **2.63~3.03** | **0.95~0.99** | **2.39~2.67** | **3.84~4.11** | **10.30~11.22** | **-1.66~-0.89** |

## 5. Conclusions

Applying optimal water allocation models to simultaneously enable economic benefits,

water preferences and environmental demands at different decision levels, time scales

and regions is a challenge. In this study, a new process-based three-layer synergistic

optimal allocation model (PTSOA) is developed and applied to a real and complex

water allocation system. The objective functions were divided into three layers to

coordinate conflicts of interest among decision makers at different levels and time

scales. Furthermore, the allocation of reclaimed water was embedded in the proposed

model for synergistic optimal allocation of both conventional and unconventional water.

A synergistic index based on network analysis was introduced to reduce competition

among different stakeholders and facilitate the positive effect of stakeholder

interactions. A hierarchical optimal algorithm was designed to solve the PTSAO model.

The proposed model was applied to a representative city in Southeast China with

scarce water resources and a developed industry. Achieving the optimal allocation of

water resources in this kind of highly developed area offers a valuable reference for

other counties in China. Key findings can be concluded from these results, as follows.

First, the results demonstrated that the PTSOA model achieved synergistic allocation





among hierarchical decision-makers across various time scales and in different regions,
yielding the highest TSI (-1.66 to -0.89) among the models evaluated. Second, with a
synergistic approach, a reasonable amount of conventional water is retained for future
use in cases with potentially high risk, with volumes of $3.95\times10^7$ m³, $3.12\times10^7$ m³, and
$2.43\times10^7$ m³ retained in normal, dry and extremely dry scenarios, respectively.
Moreover, $7.35\times10^7$ m³, $7.56\times10^7$ m³, and $7.37\times10^7$ m³ of conventional water is saved
in the three scenarios. Third, considering both reclaimed water and conventional water
in the optimization process efficiently improves the quality of municipal water, and
more than 1272.21 t/year and 48.81 t/year of COD and ammonia nitrogen emissions
are mitigated compared to those in the current situation. Distinct from previous models,
the proposed optimal model was implemented with the consideration of spatial
dimensions, which are important but often neglected. The results show that spatial
allocation yields an improvement of 4-95% for the comprehensive benefits in different
subregions compared to the benefits achieved with traditional models, and the total
comprehensive benefit increases by $1.76\times10^9$-$15.67\times10^9$ yuan compared to that in the
current situation. The synergy index established based on network analysis is used to
alleviate the competition among regions and facilitate water supply improvements.
These results and conclusions provide valuable references for the evaluations of other
complicated water allocation systems. The optimal allocation scheme is determined for
a complex water system upon consideration stakeholder synergy and various
hierarchical decision levels, time scales and regions. More in-depth studies of

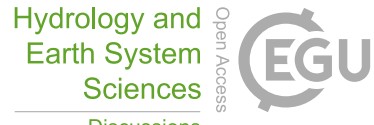

synergistic optimal water allocation are needed in the future.

*Data availability.*    The data used to support the findings of this study are available
from the corresponding author upon request.

*Author contribution.*    JL and YPX designed all the experiments. JL and SWW collected
and preprocessed the data. JL and WZ conducted all the experiments and analysed the
results. JL wrote the first draft of the manuscript with contributions from SWC. YPX
supervised the study and edited the manuscript.

*Competing interests.*    At least one of the (co-)authors is a member of the editorial
board of Hydrology and Earth System Sciences.

*Disclaimer.* Publisher's note: Copernicus Publications remains neutral with regard to
jurisdictional claims in published maps and institutional affiliations.

*Acknowledgements.*    The editors and two reviewers are greatly acknowledged for their
constructive comments to improve the quality of this paper. The Water Resources
Department of Zhejiang Province and the Yiwu City Water Construction Group Co.,
Ltd., are greatly acknowledged for providing the data regarding the water system of
Yiwu city used in this study.

*Financial support.*    This research is funded by the Major Project of Zhejiang Natural
Science Foundation (LZ20E090001) and the Zhejiang Key Research and Development
Plan (2021C03017).





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
