# Peer review of "Process-based three-layer synergistic optimal allocation model for complex water resource systems considering reclaimed water"

_Hydrology and Earth System Sciences, 2023_

## Author Comment (AC1)

**Process-based three-layer synergistic optimal allocation model for complex water resource systems considering reclaimed water**

**(hess-2023-160)**

The authors deeply appreciate the editors and reviewers for providing constructive suggestions and valuable comments, as well as positive feedback. Your suggestions will be of great help to improve the quality of our manuscript. Changes made in the revised manuscript are marked with track changes. Responses are made to all the comments and suggestions raised by the associate editor and reviewers, and are briefly described as follows.

**Response to Reviewer #1's comments:**

*1.Line 225-226: is there any other index can be used for comparison? How to evaluate the validity of this proposed index?.*

Thanks for the comment! It is really beneficial for the improvement of our paper. As for the first sub-advice that "*is there any other index can be used for comparison*", the index for comparision has been supplemented in Lines 227-230 as follows: "System entropy (H(S)) can describe the evolution direction of a water resource system and was used to promote the coordination of water supply departments in a water resource allocation system(Li et al., 2022). So, it was used for comparison to evaluate the validity of this proposed index." More detail has been added in line 608 as follows:"This factor is also used to be compared with proposed index."

For the second sub-advice that *How to evaluate the validity of this proposed index"*, we have discussed and proved it in Lines 834-842 as follows : "In Fig.6, the value of TSI are significant diverse among different scenarios as well as different solutions. As a contrast, the value of H, which is used for comparison and construction of TSI, show slight difference among solutions and even are the same in some classes. Therefore, it is difficult for decision makers to select the best solution among all candidates if we only use H for evaluation and selection in the decision process. Compared to H, TSI

introduce SSI into evaluation and the difference of coordination relationship between different schemes is distinguished by SSI. But H only pay attention to the equity among the stakeholders. So, TSI is more effective and validity than H in some extent."

*2. The titles of some sections need to be revised to make the structure of the manuscript much clearer: for example, there is "2.1.2 Constraints" and "2.2.2 Constraints". And some other sections also have this kind of issue.*

Thanks for giving the useful suggestion. To clear the structure of the manuscript, the titles of some sections are revised. For example: "2.1.1 Objective functions of the first layer; 2.1.2 Constraints of the first layer; 2.2.1 Objective functions of the second layer; 2.2.2 Constraints of the second layer; 2.3.1 Objective function of the third layer; 2.3.2 Constraints of the third layer."

*3.Fig 3. It seems that the river network is not well connected. What is the reason for that?*

Thanks for the question. In the Fig.3, the light blue lines represent river network, and the white dash-dotted lines are boundaries of different sub-regions. We are not sure what you said is the blue lines or white lines. If what you said is the blue ones, we could see that most rivers ate connected in the Fig.3. Because Yiwu City have two river systems, one is Yiwu River system with Yiwu River as the main stream, and another is Pujiang River system with Pujiang as the main stream. However, only fewer tributaries of the later one flow through Yiwu City, so it seems that the river network is not well connected in the west frontier of the Yiwu City. If what you said is the white ones, it may be mainly because the legend of the Fig.3 is not clear enough. In case of unnecessary misunderstanding, the Fig.3 has been revised as the following:

[Figure]

Fig. 3. Map of the study area

*4. Some references need to be added to prove the choice or statement in the manuscript, such as 1) Line 699-700: agricultural irrigation water, which accounts for only a small portion of the total water demand in the area: is there any official report or reference to support this statement? 2) Line 760-761: why COD, TP and TN are selected?*

Thanks for giving the comment. It's helpful for improvement of the paper.

1) The official report to support the statement of agriculture irrigation water has been added as the reference in the line 709-710 as the following: "Since there are no data available for agricultural irrigation water, which accounts for only a small portion of the total water demand in the area, and most agricultural irrigation water is supplied from surface water stored in hundreds of small reservoirs and mountain ponds (2020 Yiwu Ecological Environment Status Bulletin)".

2) It is necessary to explain why these three factors are selected. In the line 770-772, the reason has been explained as follows: "COD, TP and TN are major pollutants in Yiwu City (Yiwu Ecological City Construction Plan), and they are also major controlled pollutants of all the monitoring sections. So, these there were selected as representative pollutants in the tributaries to guarantee the water environmental quality of inland rivers."

*5. Line 745-746: How is the reduction coefficient k identified? What is the value?*

Thanks for your comment. The reduction coefficient k is identified by expert experience, and the value is 0.88. The related statement has been supplemented in the line 755-758: "the difference of this coefficient is quite slight within a small watershed (Zhao, 2014). Thus, $k$ is simplified to the same value 0.88 is the same for every reservoir and varies throughout the year according to expert experience"

*6. Probably you can delete the "Results of the" in the title of section 4.1 and some other sections.*

Thanks for your comment!
All the "Results of the" have been deleted in the manuscript.

*7. Please revise the "yuan" as "Chinese Yuan" or "Renminbi"*

Thanks for the comment! It is important to use standardized unit expressions. All the "yuan" have been revised as Chinese Yuan. There are 32 substitutions in total.

*8. There are some typo errors in the manuscript, for example Line 765 "-~". Please check the manuscript carefully to avoid this kind of issue.*

Thanks very much for giving this helpful comment. These kind of type errors have been checked cross the paper, and all the expression of the number interval is unified as the symbol "~". There are 11 substitutions in total.

*9. The descriptions of the Application and Results and Discussion section are too tedious, so it is suggested to simplify the expression appropriately.*

Thanks very much for giving this helpful comment. The Application and Results and Discussion section have been revised and the tedious sentences have been delated or simplified. For example: sentences in line 706-708 "Additionally, excluding water from reservoirs, most agricultural irrigation water is supplied from surface water stored in hundreds of small reservoirs and mountain ponds."; sentence in line 785-788

"According to the "Yiwu Water Resources Bulletin 2020", the urban comprehensive domestic sewage quota is set to 90%, and the sewage treatment rate is set to 100%. The benefits per unit water supply for different users in different subregions are determined from the Yiwu Water Price Adjustment Plan 2020."; sentence in line 801-803 "The optimization using the Pareto concept allows the operator to choose an appropriate solution depending on the prevailing circumstances and analyse the trade-off among the conflicting objectives."; sentence in line 914-917 "In other words, the balancing of the two objectives is beneficial for managers to determine an equilibrium solution that satisfies the relevant demand and successfully avoids surplus conventional or unconventional water supply in terms of sustainable development."; sentence in line 926-928 "After selecting the three scenarios that yield the best synergy and the two best objective functions for characterizing all Pareto fronts of the second layer in each scenario, these 3×3 solutions are input to the third layer for further optimization."; sentence in 950-954 "However, the various subregions obtain the greatest benefits when maximizing the unconventional water supply in dry and extreme scenarios. This result indicates that increasing the use of unconventional water in dry and extremely dry years would significantly increase the potential benefits." and some other sentences haven been delated or simplified.

*10. In the discussion, I noticed that there are some descriptions about the complex network analysis, but these discussions are somewhat superficial. So, please highlight the role of complex network analysis in this model.*

Thanks for the comment!

The role of complex network analysis have been highlighted in line 983-985 "Complex network analysis help reveal the interactions among different objectives, we determine the level of synergy in complicated water systems, identify the challenges and opportunities for sustainable development of water systems in cities with various subregions, and provide valuable insights and specific action priorities for these regions."

*11. Some conclusions should be more organized and distinct..*

Thanks for the comment! The main section of conclusions have been reorganized as follows: "The proposed model was applied to a representative city in Southeast China with scarce water resources and a developed industry. Achieving the optimal allocation of water resources in this kind of water-scarce city offers a valuable reference for other counties in China. Key advantages of PTSOA can be concluded from these results, as follows. Firstly, the results demonstrated that the PTSOA model achieved synergistic allocation among hierarchical decision-makers across various time scales and in different regions, yielding the highest TSI (-1.66 to -0.89) among the contrast models evaluated. Secondly, with a synergistic approach, a reasonable amount of conventional water is retained for future use in cases with potentially high risk, with volumes of $3.95 \times 10^7$ m³, $3.12 \times 10^7$ m³, and $2.43 \times 10^7$ m³ retained in normal, dry and extremely dry scenarios, respectively. Moreover, $7.35 \times 10^7$ m³, $7.56 \times 10^7$ m³, and $7.37 \times 10^7$ m³ of conventional water is saved in the three scenarios. Thirdly, considering both reclaimed water and conventional water in the optimization process efficiently improves the quality of municipal water, and more than 1272.21 t/year and 48.81 t/year of COD and ammonia nitrogen emissions are mitigated compared to those in the current situation. Lastly, distinct from previous models, the proposed optimal model was implemented with the consideration of spatial dimensions, which are important but often neglected. The results show that spatial allocation yields an improvement of 4-~95% for the comprehensive benefits in different subregions compared to the benefits achieved with traditional models, and the total comprehensive benefit increases by $1.76 \times 10^9$-~$15.67 \times 10^9$ Chinese Yuan compared to that in the current situation."

*compared to current situation." However, I didn't find other supporting material in the manuscript, so please clarify it.*

Thanks for the comment!

For the first part "the total amount of conventional water is saved, which is $7.35 \times 10^7$ m³, $7.56 \times 10^7$ m³, $7.37 \times 10^7$ m³ in the scenarios, respectively." The supporting explanation has been added in line 709-713 as follows "Moreover, by selecting the solution with highest TSI, $7.35 \times 10^7$ m³, $7.56 \times 10^7$ m³, and $7.37 \times 10^7$ m³ of unconventional water would be supplied as an effective supplement to conventional water. In the other word, conventional water would be saved by our proposed model and index in the three scenarios."

For the second part "Thirdly, engaging both reclaimed water and conventional water in the process of optimization efficiently improves the municipal water environmental quality, and more than 1272.21t/year and 48.81t/year emissions of COD and ammonia nitrogen are reduced compared to current situation." The supporting explanation is in line 709-713 as follows "Additionally, based on the constraints regarding the contaminants allowed to be discharged, more than 1272.21 t and 48.81 t of COD and ammonia nitrogen emissions are avoided per year."

*13. What are the main influencing factors of the proposed model? Although this manuscript gives many indices of the model, it is difficult to know the main influential factors of the PTSOA model. Please clarify it in the manuscript.*

Thanks very much for giving this helpful comment.

The proposed PTSOA model is influenced by many factors. The main factors are listed in the manuscript. However, it seems also hard to identify which one is more important. So, the statement of main influential factors has been added in line 785-787 "There are plenty of influencing factors in the model, the most important ones among them are the value of water demand, the value of available water and some key hyper-parameter."

*14. What is the specific meaning of the "complex water resources system" in the title? In case of misunderstanding, please define it clearly in the manuscript.*

Thanks very much for giving this helpful comment. It is necessary to define the key words in the title across the manuscript. The specific meaning of the complex water resources system has been added in line 38-41 "Nowadays, the water resources system become more and more complex, and is consisted with multiple sources and users as well as water reused infrastructure. This kind of water resources system is called complex water resources system in the following." Also, some other expression about complex water resources system have also been unified in the manuscript.

*15.  Some abbreviations are repeatedly explained in the manuscript. For example, Line 235: "a new reasonable evaluation index named synergy index of the system (TSI)", and Line 611 says "synergy index of the system (TSI) is used for…". Please check all repeats through the manuscript.*

Thanks very much for giving this helpful comment. All repeats through the manuscript about "synergy index of the system (TSI)" have been checked and revised. There are a total of six amendments

*16.  Some units have no brackets, but some do. For example, in Table 2, all units don't have brackets, but the units in Table 3 have. Please adjust them to journal format requirements.*

Thanks very much for giving this helpful comment. According to journal format requirements, all the units in the tables haven been added brackets. Table 1-3 are unified for standard.

*17.  The fonts in Fig.4 are not vary clear. Maybe it is because of the color and size of the fonts. Please adjust them for easier reading.*

Thanks very much for giving this helpful comment. For easier reading of readers, Fig.4 has been repainted as following:

[Figure]

**Fig. 4. Schematic diagram of Yiwu city**

---

## Author Comment (AC2)

**Response to Reviewer #2's comments:**

*1. Line 305 – 320*

*In the equation 10 and 13, for calculating maximum allowable storage capacity, there is a precipitation component associated with the water source. I'm uncertain about the methodology used to quantify this. Is a rainfall runoff model employed for this purpose?*

Thanks for your question. Well, the principle of equitation 10 and 13 is based on hydrologic budget and conservation of mass. In the equitation, the precipitation component associated with the water source could be obtained by observed data of each sub-catchment extracting of the water source or rainfall runoff model as you said. The methodology depends on the data available in the study area. In our application of the model, this precipitation component associated with the water sources were calculated by Tyson polygon method based on the measured data of seven rainfall stations (Shi Caotou, Suxi, Yiwu, Fotang, Baifeng, Fengkeng, Changfu) in the basin in normal (1984.1–1985.1), dry (2008.1–2009.1), and extremely dry (1971.1–1972.1) scenarios, relatively. Limited by the space constraints, this section does not elaborate. Thanks for your question reminding us the importance to clarify the issue. So, the related supplementary statement has been added in line 771-776: "In our application of the model, this precipitation component associated with the water sources were calculated by Tyson polygon method (Liu et al., 2014) based on the measured data of seven rainfall stations (Shi Caotou, Suxi, Yiwu, Fotang, Baifeng, Fengkeng, Changfu) in the basin in normal (1984.1–1985.1), dry (2008.1–2009.1), and extremely dry (1971.1–1972.1) scenarios, severally."

*2. Line 601– 679*

*The introduction of study area doesn't clearly indicate whether the reservoir serves an*

*energy-related purpose. If it does serve an energy-related purpose, it's unclear whether the impact on energy production has been considered in an analysis.*

Thanks for your question. The energy-related purpose is quite important part for reservoir operation which has generating function. However, the seven main reservoirs do not have generation function in Yiwu City. Because the main power generation mode of Yiwu city is photovoltaic power generation and others (Yiwu City government service portal), and the reservoirs do not need to undertake generating function limited by relatively low elevation difference in this area. So, the impact on energy production has not been considered in an analysis. But your consideration is really enlightening, thus we will focus on the reservoirs serves energy-related purpose in other study area and further enrich the model in the future study.

*3.    Line 609– 612*

*The three indices introduced here require further elaboration to help readers interpret the case study results. For instance, it would be beneficial to explain the range of values for these indices and what high values, such as H and others, signify. Additionally, insights into what lower values indicate would be valuable.*

Thanks for your valuable comment! The introduction of the three core indices is a little rough in the previous manuscript. We deeply agree with your comment and suggestion. So, the range of values for these indices and what high values as well as what lower values indicate have been classified in line 631-639: "SSI is ranged form 0~N, and higher SSI indicates higher connectivity of the objects in the system which means they are easier to promote each other, and lower SSI indicates lower connectivity which means the promotion is hard to realize and obstacles to each other may occur. H is ranged from 0~N*log(1/N), lower H indicates better overall equilibrium and higher H

indicates worse overall equilibrium from objective perspective. TSI is greater than 0. When a water resource system's TSI value is higher, the degree of synergy is higher; conversely, when a water resource system's degree of synergy is lower, the TSI value is lower. In our application, based on actual evaluation, the criteria as divided. When TSI$\geq$5, the degree of sunergy is considered satisfactory. Additionally, 5>TSI$\geq$3 is moderate synergy degree and 3>TSI is low degree."

*4.   Line 775– 775*

*To provide a more comprehensive overview of the optimization process, it would be beneficial to include information about the computational setup and the time required for the analysis. For instance, it would be helpful to know how long it took to generate Pareto sets across 500 runs of the PTSOA model and whether high-performance computing was utilized.*

Thanks for your valuable comment! It is quite beneficial to include information about the computational setup and the time required for the analysis. There are 1000 iterations of each run in most cases. If The feasible solutions could not be found in some cases, the number of iteration would be increased. Based on the log recording, this important information had been added in line 819-823 of the revised manuscript: "If The feasible solutions could not be found in some cases, the number of iteration would be increased. It took approximately mean 17 h of CPU time on a computer with 32 GB memory and intel corei7@3.4 GHz of CPU. Therefore, in this study, each iteration for a single trial solution takes 0.12 s of CPU time on the computer with the named specifications."

*5.   Line 808-809 and 889 - 891*

*The labels on Figure 5 and Figure 9 are nearly impossible to read, even when I zoom in to view the names of the classes. Please consider using different label colors and adjusting the background to ensure the labels are easily discernible.   Additionally, please explain the Figure labels (F1, F2, F3) in the caption.*

Thanks for your comment! The Figure 5 and Figure 9 have been repainted by using different label colors and enlarging the sub-figures. Hope they are easier to read now.

Additionlly, the labels in Figure 1, Figure 2 and Figure 3 have been explained in the caption in line 208-212, line 235-238, line 709-712 as follows: "In Fig. 1, the grey boxes indicatethe three different allocation dimensions, the green boxes indicates the three different decision levels coupled with spatial scales, the bright yellow boxes indicates every key nodes in the whole allocation process and the buff boxes indicates nested time scale." "In Fig. 2, there are three layers in the framework and each layer has two parts: multi-objective optimal water resources allocation and collaborative water resources allocation for objectives. In the multi-objective optimal water resources allocation sub-layers contain key nodes in the allocation process and relevant objectives and constraints. In the collaborative water resources allocation for objectives sub-layers contain optimization algorithm and decision selection method." "In Fig. 3, the white labels indicate five sub-regions in the city, the black labels near the reservoirs are their name, the black labels named O1~O6 indicates the name of the water distribution outlets and the labels near the lifting pump station are their names."

*6.Line 798 - 800*

*Could you please clarify how the selected decision alternatives achieve a water supply reliability greater than 95% under the three different conditions? It would be helpful to understand the approach used to derive this information from these three panel plots.*

Thanks for your question. It is necessary to clarify how the selected decision alternatives achieve a water supply reliability greater than 95% under the three different conditions. There are 6×3 decision alternatives selected in the six clusters of the optimal first-layer results. To help to understand the approach used to derive this information from these three panel plots, the clarification is added in line 358-361 of the revised manuscript as follows: "The water shortage varies in the range of $-1.2 \times 10^6 \sim 0.8 \times 10^5$ m³, $-0.5 \times 10^5 \sim 2.0 \times 10^6$ m³, $0 \sim 3.5 \times 10^6$ m³ in normal, dry and extremely dry scenarios respectively. The average water demand is around $1.8 \times 10^8$ m³, and water shortage of the selected decision alternatives are all less $9 \times 10^6$ m³." Thanks to your suggestion, this part seems clearer.

*8. Line 1006- 1008*

*The performance of the PTSOA model is compared with some known MOEAs. Yet there*

*are other algorithms that perform better than the ones that have been tested. For example, Borg MOEA has accomplished superior performance levels across a wide number of challenging multi-objective problems by meeting or exceeding the performance of other state-of-the-art MOEAs. It would be interesting to test the Borg algorithm as well to see if it can produce different results. It would also be valuable to compare the computational time of these MOEAs with the time required for your model.*

Thanks for your comment! Borg Multi-objective Evolutionary Algorithm (MOEA), an efficient and robust many-objective optimization tool. It is characterized by its use of auto-adaptive multi-operator search and other adaptive features, allowing the algorithm to tailor itself to local search conditions encountered during optimization. Using a rigorous diagnostic framework, the Borg MOEA is distinguished against a broad sample of state-of-the-art MOEAs. The Borg MOEA meets or exceeds the efficiency, reliability, and search quality of other MOEAs on the majority of many problems (David M. Hadka,2013). Large-scale parallelization of the Borg MOEA for many objective optimization of complex environmental systems. The multimaster Borg MOEA (Hadka and Reed, submitted manuscript, 2014) combines two parallelization paradigms: (1) master-worker distributed function evaluations and (2) multiple cooperating search populations (also termed the island model [Cantu-Paz, 2000]). Effective parallelization of the multimaster Borg MOEA maximizes this parameter for a given amount of wall-clock time. So, multimaster Borg MOEA seems quite suitable for many-objective optimization of the complex system.

Based on your helpful comment, the Borg MOEA has been tested and compared with other algorithms. In the TSI dimension, it is performance is slightly worse as shown in Table 4 of the revised version. In this study, our main focus is to find the most collaborative solution through optimization. Thus, PTSOA has accomplished superior performance level in this respect. However, we are surprised to find that the Borg MOEA algorithm could save around one-fifth of the computing time of the model. So, in the future, we may be interested in figure out how to coupling the Borg MOEA algorithm with our PTSOA model in a more efficient and synergetic way. The replenishment about the Borg MOEA has been added in line 1128-1137 as follows:

"Borg MOEA, an efficient and robust many-objective optimization tool. It is characterized by its use of auto-adaptive multi-operator search and other adaptive features (Reed et al., 2013). The TSI of Borg MOEA is lower than PTSOA. Therefore, in the TSI dimension, it is performance is slightly worse than PTSOA model. However, it is noticed that the Borg MOEA algorithm could save around one-fifth of the computing time of the model (around 7h). In the future, it would be interesting to figure out how to coupling the Borg MOEA algorithm with our PTSOA model in a more efficient and synergetic way. In this study, our main focus is to find the most synergetic solution through optimization. Thus, PTSOA has accomplished superior performance level in this respect."

*9. A general issue:*

*Each figure and table in the paper must have a caption that provides enough information that a reader can understand the data presented without referring to the text.*

Thanks for your comment! This comment is really valuable to improve this paper. Most figures and tables have been completed by a caption providing enough information. The captions are modified as follows:

"Fig. 5. Sets of Pareto solutions after 500 model simulations with the hierarchical optimal algorithm under (a) normal, (b) dry and (c) extremely dry scenarios. (F1: total water supply shortage, 104m3; F2: total water supply benefit, 104 Chinese Yuan; F3: the total amount of reserved water in reservoirs, 104m3. The red arrow indicates the direction of optimization. K1-n,K2-n and K3-n represents the nth class of solutions in the normal, dry and extremely dry scenario separately, n=1~6.)"

"Fig. 6. Comparison of TSI (total synergy index), SSI (total connectivity) and H (overall equilibrium) values among various Pareto solutions in different classes for the (K1) normal, (K2) dry, and (K3) extremely dry scenarios. (K1-n,K2-n and K3-n represents the nth class of solutions in the normal, dry and extremely dry scenario separately, n=1~6.)"

"Fig. 7. Water supply from each reservoir to connected water works in each month in the normal scenario 104 m3

(K1-n represents the nth class of solutions in the normal scenario, n=1~6.)"

"Fig. 8. Pareto fronts of the second layer in the PTSOA model after 500 simulations with the hierarchical optimal algorithm in the normal, dry and extremely dry scenarios. (F1 represents the total amount of water retained in water works ,$10^4m^3$; F2 represents the amount of unconventional water supplied,$10^4 m^3$. The direction of optimization is from the top-right corner to the bottom-left corner. K1-n represents the nth class of solutions in the normal scenario, K2-n represents the nth class of solutions in the dry scenario, and K3-n represents the nth class of solutions in the extremely dry scenario, n=1~6.)"

"Fig. 9. Illustration of parallel-reference Pareto sets from the third layer in the PTSPOA model attained across all runs for the (S1) normal, (S2) dry, and (S3) extremely dry scenarios (S1-1 represents the normal scenario with the minimum total amount of water retained in water works, S1-2 represents the normal scenario with the maximum unconventional water supply and S1-3 represents the normal scenario with the maximum synergy degree in the second layer)"

"Fig.10. Comprehensive benefit in five sub-regions after the regional collaborative allocation of water resources (S1 represents normal scenario, S2 represents dry scenario, and S3 represents extremely dry scenarios; S1-1 represents the normal scenario with the minimum total amount of water retained in water works, S1-2 represents the normal scenario with the maximum unconventional water supply and S1-3 represents the normal scenario with the maximum synergy degree in the second layer)"

*10.Line 251, ...*

*Each section that describes the three layers of the process shares the same subsection name; I would recommend renaming them to avoid any confusion.*

Thanks for your comment! The names of the sub-sections have been corrected in the revised version. For example: 2.1.1Objective functions of the first layer; 2.1.2 Constraints of the first layer; 2.2.1 Objective functions of the second layer; 2.2.2 Constraints of the second layer; 2.3.1  Objective function of the third layer; 2.3.2 Constraints of the third layer.

*11. Line 228: space after "interactions"*

Thanks for your comment! The whole paper has been checked and the missing spaces have been added like the space after "interactions".

---

## Author Response (AR1)

**Process-based three-layer synergistic optimal allocation model for complex water resource systems considering reclaimed water**

**(hess-2023-160)**

The authors deeply appreciate the editors and reviewers for providing constructive suggestions and valuable comments, as well as positive feedback. Your suggestions will be of great help to improve the quality of our manuscript. We have revised our manuscript based on your comments. Changes made in the revised manuscript are marked with tracked changes. Responses are made to all the comments and suggestions raised by the associate editor and reviewers, and are briefly described as follows.

**Response to Reviewer #1's comments:**

*1.Line 225-226: is there any other index can be used for comparison? How to evaluate the validity of this proposed index?.*

Thanks for the comment! It is really beneficial for the improvement of our paper. As for the first sub-advice that "*is there any other index can be used for comparison",* the index for comparison has been supplemented in Lines 227-230 as follows: "System entropy (H(S)) can describe the evolution direction of a water resource system and was used to promote the coordination of water supply departments in a water resource allocation system (Li et al., 2022). So, it was used for comparison to evaluate the validity of this proposed index." More detail has been added in Line 608 as follows: "This factor is also used to be compared with the proposed index."

For the second sub-advice that "*How to evaluate the validity of this proposed index*", besides the use of system entropy (*H*), we have discussed and proved it in Lines 903-911 as follows : " In Fig.6, the value of *TSI* are significantly diverse among different scenarios as well as different solutions. *H* is widely used to evaluate the equality of different solutions (Gao et al., 2013;Li et al., 2022). As a contrast, the value of *H*, which is used for comparison and construction of *TSI*, show slight differences among solutions and even are the same in some classes. Therefore, it is difficult for decision makers to

select the best solution among all candidates if we only use *H* for evaluation and selection in the decision process. Compared to *H*, *TSI* introduces *SSI* into evaluation and the difference of coordination relationship between different schemes is distinguished by *SSI*. But *H* only pays attention to the equity among the stakeholders. So, *TSI* is more effective and valid than *H* in some extent."

*2.    The titles of some sections need to be revised to make the structure of the manuscript much clearer: for example, there is "2.1.2 Constraints" and "2.2.2 Constraints". And some other sections also have this kind of issue.*

Thanks for giving the useful suggestion. To clarify the structure of the manuscript, the titles of some sections are revised. For example: "2.1.1 Objective functions of the first layer; 2.1.2 Constraints of the first layer; 2.2.1  Objective functions of the second layer; 2.2.2 Constraints of the second layer; 2.3.1 Objective function of the third layer; 2.3.2    Constraints of the third layer."

*3.Fig 3. It seems that the river network is not well connected. What is the reason for that?*

Thanks for the question. In Fig.3, the light blue lines represent river network, and the white dash-dotted lines are boundaries of different sub-regions. From this figure we could see that most rivers are connected but some not. Yiwu City has two river systems. One is Yiwu River system with Yiwu River as the main stream, and another is Pujiang River system with Pujiang as the main stream. However, only limited tributaries of the later flow through Yiwu City, so the river network is not well connected in the frontier of the Yiwu City. In case of unnecessary misunderstanding (with diashed-dotted lines), Fig.3 has been revised as the following:

[Figure]

Fig. 3. Map of the study area

*4. Some references need to be added to prove the choice or statement in the manuscript, such as 1) Line 699-700: agricultural irrigation water, which accounts for only a small portion of the total water demand in the area: is there any official report or reference to support this statement? 2) Line 760-761: why COD, TP and TN are selected?*

Thanks for the nice comments.

1) The official report to support the statement of agriculture irrigation water has been added as the reference in the line 762-766 as the following: "There is no data available for agricultural irrigation water, and it only accounts for a small portion of the total water demand in the area, as well as most agricultural irrigation water is supplied from surface water stored in hundreds of small reservoirs and mountain ponds (Yiwu Ecological Environment Status Bulletin, 2020)".

2) It is necessary to explain why these three factors are selected. In the line 837-839, the reason has been given as follows: "the results are listed in Table 3. COD, TP and TN are major pollutants in Yiwu City (Yiwu Ecological Environment Status Bulletin, 2020), and they are also major controlled pollutants of all the monitoring sections. So, these were selected as representative pollutants in the tributaries to guarantee the water environmental quality of inland rivers."

5.   Line 745-746: How is the reduction coefficient k identified? What is the value?

Thanks for your comment. The reduction coefficient k is identified by expert experience, and the value is 0.88. The related statement has been supplemented in the line 816-818: "the difference of this coefficient is quite slight within a small watershed (Zhao, 2014). Thus, $k$ is simplified to the same value 0.88 and is the same for every reservoir and varies throughout the year according to expert experience(Zhao, 2014)."

6.   Probably you can delete the "Results of the" in the title of section 4.1 and some other sections.

Thanks for your comment!
All the "Results of the" have been deleted in the manuscript.

7.   Please revise the "yuan" as "Chinese Yuan" or "Renminbi"

Thanks for the comment! It is important to use standardized unit expressions. All the "yuan" have been revised as Chinese Yuan. There are 32 substitutions in total.

8.   There are some typo errors in the manuscript, for example Line 765 "-~". Please check the manuscript carefully to avoid this kind of issue.

Thanks very much for giving this helpful comment. These kinds of type errors have been checked cross the manuscript, and all the expression of the number interval is unified as the symbol "~". There are 11 substitutions in total.

9.   The descriptions of the Application and Results and Discussion section are too tedious, so it is suggested to simplify the expression appropriately.

Thanks very much for giving this helpful comment. The Application and Results and Discussion section have been revised and the tedious sentences have been deleted or simplified. For example: sentences in line 743-744 "Additionally, most agricultural irrigation water is supplied from surface water stored in hundreds of small reservoirs and mountain ponds."; sentence in line 785-788 "According to Yiwu Water Resources

Bulletin (2020), the urban comprehensive domestic sewage quota is set to 90%, and the sewage treatment rate is set to 100%. The benefits per unit water supply for different users in different subregions are determined from the Yiwu Water Price Adjustment Plan 2020."; sentence in line 801-803 "The optimization using the Pareto concept allows the operator to choose an appropriate solution depending on the prevailing circumstances and analyze the trade-offs among the conflicting objectives."; sentence in line 914-917 "In other words, the balancing of the two objectives is beneficial for managers to determine an equilibrium solution that satisfies the relevant demand and successfully avoids surplus conventional or unconventional water supply in terms of sustainable development."; sentence in line 926-928 "After selecting the three scenarios that yield the best synergy and the two best objective functions for characterizing all Pareto fronts of the second layer in each scenario, these 3×3 solutions are input to the third layer for further optimization."; sentence in 950-954 "However, the various subregions obtain the greatest benefits when maximizing the unconventional water supply in dry and extreme scenarios. This result indicates that increasing the use of unconventional water in dry and extremely dry years would significantly increase the potential benefits." and some other sentences haven been deleted or simplified.

*10. In the discussion, I noticed that there are some descriptions about the complex network analysis, but these discussions are somewhat superficial. So, please highlight the role of complex network analysis in this model.*

Thanks for the comment!

The role of complex network analysis has been highlighted in lines 1076-1081 "Complex network analysis helps reveal the interactions among three layers with different dimensions. We determine the level of synergy in complicated water systems, identify the challenges and opportunities for sustainable development of water systems in cities with various sub-regions, and provide valuable insights and specific action priorities for these regions."

*11. Some conclusions should be more organized and distinct..*

Thanks for the comment! The main section of conclusions has been reorganized as follows: "The proposed model was applied to a city in Southeast China with scarce water resources and developed industry. Achieving the optimal allocation of water resources in this kind of water-scarce city offers a valuable reference for other counties in China. The key findings of this study are as follows. Firstly, the results demonstrated that the PTSOA model achieved synergistic allocation among hierarchical decision-makers across various time scales and in different regions, yielding the highest *TSI* (-1.66 to -0.89) among the contrast models evaluated. Secondly, with a synergistic approach, a reasonable amount of conventional water is retained for future use in cases with potentially high risk, with volumes of $3.95 \times 10^7$ m³, $3.12 \times 10^7$ m³, and $2.43 \times 10^7$ m³ retained in normal, dry and extremely dry scenarios, respectively. Moreover, $7.35 \times 10^7$ m³, $7.56 \times 10^7$ m³, and $7.37 \times 10^7$ m³ of conventional water can be saved in the three scenarios. Thirdly, considering both reclaimed water and conventional water in the optimization process efficiently improves the quality of municipal water, and more than 1272.21 t/year and 48.81 t/year of COD and ammonia nitrogen emissions are mitigated compared to those in the current situation. Lastly, distinct from previous models, the proposed optimal model was implemented with consideration of spatial dimensions, which are important but often neglected. The results show that spatial allocation yields an improvement of 4-~95% for the comprehensive benefits in different subregions compared to the benefits achieved with traditional models, and the total comprehensive benefit increases by $1.76 \times 10^9$-~$15.67 \times 10^9$ Chinese Yuan compared to that in the current situation."

*12. In the conclusion, attention should be paid to the results derived in this study. For example, Line 1069-1073: "the total amount of conventional water is saved, which is $7.35 \times 10^7$ m³, $7.56 \times 10^7$ m³, $7.37 \times 10^7$ m³ in the scenarios, respectively. Thirdly, engaging both reclaimed water and conventional water in the process of optimization efficiently improves the municipal water environmental quality, and more than 1272.21t/year and 48.81t/year emissions of COD and ammonia nitrogen are reduced compared to current situation." However, I didn't find other supporting material in the*

*manuscript, so please clarify it.*

Thanks for the comment!

For the first part "the total amount of conventional water is saved, which is $7.35 \times 10^7$ m³, $7.56 \times 10^7$ m³, $7.37 \times 10^7$ m³ in the scenarios, respectively." The supporting explanation has been added in line 709-713 as follows "Moreover, by selecting the solution with highest TSI, $7.35 \times 10^7$ m³, $7.56 \times 10^7$ m³, and $7.37 \times 10^7$ m³ of unconventional water would be supplied as an effective supplement to conventional water. In the other word, conventional water would be saved by our proposed model and index in the three scenarios."

For the second part "Thirdly, engaging both reclaimed water and conventional water in the process of optimization efficiently improves the municipal water environmental quality, and more than 1272.21t/year and 48.81t/year emissions of COD and ammonia nitrogen are reduced compared to current situation." The supporting explanation is in line 709-713 as follows "Additionally, based on the constraints regarding the contaminants allowed to be discharged, more than 1272.21 t and 48.81 t of COD and ammonia nitrogen emissions are avoided per year."

*13. What are the main influencing factors of the proposed model? Although this manuscript gives many indices of the model, it is difficult to know the main influential factors of the PTSOA model. Please clarify it in the manuscript.*

Thanks very much for giving this helpful comment.

The proposed PTSOA model is influenced by many factors. The main factors are listed in the manuscript. However, it seems also hard to identify which one is more important. So, the statement of main influential factors has been added in lines 785-787 "There are many influencing factors in the model and the most important ones among them are water demand, available water and key hyper-parameters."

*14. What is the specific meaning of the "complex water resources system" in the title? In case of misunderstanding, please define it clearly in the manuscript.*

Thanks very much for giving this helpful comment. It is necessary to define the key

words in the title across the manuscript. The specific meaning of the complex water resources system has been added in lines 38-41 "Nowadays, the water resources system become more and more complex, and often has multiple sources and users as well as water reused infrastructure. This kind of water resources system is called complex system in the current study.".

*15. Some abbreviations are repeatedly explained in the manuscript. For example, Line 235: "a new reasonable evaluation index named synergy index of the system (TSI)", and Line 611 says "synergy index of the system (TSI) is used for…". Please check all repeats through the manuscript.*

Thanks very much for giving this helpful comment. All repeats through the manuscript about "synergy index of the system (*TSI*)" have been checked and revised. There are a total of six amendments

*16. Some units have no brackets, but some do. For example, in Table 2, all units don't have brackets, but the units in Table 3 have. Please adjust them to journal format requirements.*

Thanks very much for giving this helpful comment. According to journal format requirements, all the units in the tables haven been added with brackets. Table 1-3 are unified.

*17. The fonts in Fig.4 are not vary clear. Maybe it is because of the color and size of the fonts. Please adjust them for easier reading.*

Thanks very much for giving this helpful comment. For easier reading of readers, Fig.4 has been repainted as following:

[Figure]

**Fig. 4. Schematic diagram of Yiwu city**

**Response to Reviewer #2's comments:**

*1.   Line 305 – 320*

*In the equation 10 and 13, for calculating maximum allowable storage capacity, there is a precipitation component associated with the water source. I'm uncertain about the methodology used to quantify this. Is a rainfall runoff model employed for this purpose?*

Thanks for your question. The principle of equation 10 and 13 is based on hydrologic budget and conservation of mass. The maximum allowable capacity is calculated based on hydrologic budget and conservation of mass. In the equation, this precipitation component associated with the water sources were calculated by the Thiessen polygon method or rainfall runoff model as you said. The methodology depends on the data available in the study area. In our application of the model, this precipitation component associated with the water sources were calculated by Thiessen polygon method based on the measured data of seven rainfall stations (Shi Caotou, Suxi, Yiwu, Fotang, Baifeng, Fengkeng, Changfu) in the basin in normal (1984.1–1985.1), dry (2008.1–2009.1), and extremely dry (1971.1–1972.1) scenarios. Limited by the space constraints, this section is not elaborated. Thanks for your question reminding us the importance to clarify the issue. So, the related supplementary statement has been added in lines 771-776: "In our application of the model, this precipitation component was calculated by the Thiessen polygon method (Liu et al., 2014) based on the measured data of seven rainfall stations (Shi Caotou, Suxi, Yiwu, Fotang, Baifeng, Fengkeng, Changfu) in the basin in normal (1984.1–1985.1), dry (2008.1–2009.1), and extremely dry (1971.1–1972.1) scenarios."

*2.   Line 601– 679*

*The introduction of study area doesn't clearly indicate whether the reservoir serves an energy-related purpose. If it does serve an energy-related purpose, it's unclear whether the impact on energy production has been considered in an analysis.*

Thanks for your question. The energy-related purpose is quite important for reservoir which has generating function. However, the seven main reservoirs do not have energy generation purposes. Because the main power generation mode of Yiwu city is photovoltaic power generation and others (Yiwu City government service portal), and the reservoirs do not need to undertake energy generation tasks, limited by relatively low elevation difference in this area. So, the impact on energy production has not been considered in an analysis.

*3.   Line 609– 612*

*The three indices introduced here require further elaboration to help readers interpret the case study results. For instance, it would be beneficial to explain the range of values for these indices and what high values, such as H and others, signify. Additionally, insights into what lower values indicate would be valuable.*

Thanks for your valuable comment! The introduction of the three core indices is slightly rough in the previous manuscript. We deeply agree with your comment and suggestion. So, the range of values for these indices and what high values as well as what lower values indicate have been clarified in lines 631-639: " *SSI* is ranged from 0~N, and higher *SSI* indicates higher connectivity of the objects in the system which means they are easier to promote each other. *H* is ranged from 0~N*log(1/N), and lower *H* indicates better overall equilibrium from objective perspective. *TSI* is greater than 0. When a water resource system's *TSI* value is higher, the degree of synergy is higher. In our

application, based on actual evaluation, we define when $TSI \geq 5$ the degree of synergy is considered satisfactory. $5 > TSI \geq 3$ is defined as moderate and $3 > TSI$ is defined as low."

*4.   Line 775– 775*

*To provide a more comprehensive overview of the optimization process, it would be beneficial to include information about the computational setup and the time required for the analysis. For instance, it would be helpful to know how long it took to generate Pareto sets across 500 runs of the PTSOA model and whether high-performance computing was utilized.*

Thanks for your valuable comment! It is quite beneficial to include information about the computational setup and the time required for the analysis. There are 1000 iterations of each run in most cases. If the feasible solutions could not be found in some cases, the number of iteration would be increased. Based on the log recording, this important information has been added in lines 839-843 of the revised manuscript: "If the feasible solutions could not be found in some cases, the number of iteration would be increased. It took approximately 34 h of CPU time on a computer with 32 GB memory and intel corei7@3.4 GHz of CPU. Therefore, in this study, each iteration for a single trial solution takes 0.24 s of CPU time on the computer with the named specifications."

*5.   Line 808-809 and 889 - 891*

*The labels on Figure 5 and Figure 9 are nearly impossible to read, even when I zoom in to view the names of the classes. Please consider using different label colors and adjusting the background to ensure the labels are easily discernible.   Additionally, please explain the Figure labels (F1, F2, F3) in the caption.*

Thanks for your comment! Figure 5 and Figure 9 have been repainted by using different label colors and the sub-figures are enlarged. Hope they are easier to read now. Additionally, the labels in Figure 1, Figure 2 and Figure 3 have been explained in the caption in lines 208-212, lines 235-238, lines 709-712 as follows: "In Fig. 1, the grey boxes indicate the three different allocation dimensions, the green boxes indicates the

three different decision levels coupled with spatial scales, the bright yellow boxes indicates every key nodes in the whole allocation process and the buff boxes indicates nested time scale." "In Fig. 2, there are three layers in the framework and each layer has two parts: multi-objective optimal water resources allocation and collaborative water resources allocation for objectives. In the multi-objective optimal water resources allocation sub-layers contain key nodes in the allocation process and relevant objectives and constraints. In the collaborative water resources allocation for objectives sub-layers contain optimization algorithm and decision selection method." "In Fig. 3, the white labels indicate five sub-regions in the city, the black labels near the reservoirs are their names, the black labels named O1~O6 indicate the names of the water distribution outlets and the labels near the lifting pump station are their names."

*6.Line 798 - 800*

*Could you please clarify how the selected decision alternatives achieve a water supply reliability greater than 95% under the three different conditions? It would be helpful to understand the approach used to derive this information from these three panel plots.*

Thanks for your question. It is necessary to clarify how the selected decision alternatives achieve a water supply reliability greater than 95% under the three different conditions. There are 6×3 decision alternatives selected in the six clusters of the optimal first-layer results. To help understand the approach used to derive this information from these three panel plots, the clarification is added in lines 864-868 of the revised manuscript as follows: "The water shortage varies in the range of $-1.2×10^6$~$0.8×10^5$ m³, $-0.5×10^5$~$2.0×10^6$ m³, $0$~$3.5×10^6$ m³ in normal, dry and extremely dry scenarios respectively. The average water demand is around $1.8×10^8$ m³, and water shortage of the selected decision alternatives are all less $9×10^6$ m³. So, the water supply reliability of the selected decision alternatives is greater than 95% under normal, dry and extremely dry conditions with the consideration of water demand."

*8. Line 1006- 1008*

*The performance of the PTSOA model is compared with some known MOEAs. Yet there*

*are other algorithms that perform better than the ones that have been tested. For example, Borg MOEA has accomplished superior performance levels across a wide number of challenging multi-objective problems by meeting or exceeding the performance of other state-of-the-art MOEAs. It would be interesting to test the Borg algorithm as well to see if it can produce different results. It would also be valuable to compare the computational time of these MOEAs with the time required for your model.*

Thanks for your comment! Borg Multi-objective Evolutionary Algorithm (MOEA)is an efficient and robust many-objective optimization tool. The Borg MOEA meets or exceeds the efficiency, reliability, and search quality of other MOEAs on the majority of many problems (David M. Hadka,2013). The multimaster Borg MOEA (Hadka and Reed, 2014) combines two parallelizationparadigms: (1) master-worker distributed function evaluations and (2) multiple cooperating search populations (also termed the island model [Cantu-Paz, 2000]). Effective parallelization of the multimaster Borg MOEA maximizes this parameter for a given amount of wall-clock time. So, multimaster Borg MOEA seems quite suitable for many-objective optimization of the complex system.

Based on your helpful comment, the Borg MOEA has been tested and compared with other algorithms in the revised manuscript. In the *TSI* dimension, its performance is slightly worse as shown in Table 4 of the revised version. In this study, our main focus is to find the most collaborative solution through optimization. Thus, PTSOA has accomplished superior performance in this respect. However, we are surprised to find that the Borg MOEA algorithm could save around one-fifth of the computing time of the model. So, in the future, we may be interested in figuring out how to couple the Borg MOEA algorithm with our PTSOA model in a more efficient and synergetic way. The replenishment about the Borg MOEA has been added in lines 1128-1137 as follows:

"Borg MOEA is an efficient and robust many-objective optimization tool. It is characterized by its use of auto-adaptive multi-operator search and other adaptive features (Reed et al., 2013). The *TSI* of Borg MOEA is lower than PTSOA. Therefore, in the *TSI* dimension, its performance is slightly worse than the PTSOA model. However, it is noticed that the Borg MOEA algorithm could save around one-fifth of

the computing time of the model (around 7h). In the future, it would be interesting to figure out how to couple the Borg MOEA algorithm with our PTSOA model in a more efficient and synergetic way. In this study, our main focus is to find the most synergetic solution through optimization. Thus, PTSOA has accomplished superior performance in this respect.＂

*9. A general issue:*

*Each figure and table in the paper must have a caption that provides enough information that a reader can understand the data presented without referring to the text.*

Thanks for your comment! Most figures and tables have been completed by a caption providing enough information. The captions are modified as follows:

＂Fig. 5. Sets of Pareto solutions after 500 model simulations with the hierarchical optimal algorithm under (a) normal, (b) dry and (c) extremely dry scenarios. (F1: total water supply shortage, $10^4 m^3$; F2: total water supply benefit, $10^4$ Chinese Yuan; F3: the total amount of reserved water in reservoirs, $10^4 m^3$. The red arrow indicates the direction of optimization. K1-n,K2-n and K3-n represents the nth class of solutions in the normal, dry and extremely dry scenario separately, n=1~6.)＂

＂Fig. 6. Comparison of *TSI* (total synergy index), *SSI* (total connectivity) and *H* (overall equilibrium) values among various Pareto solutions in different classes for the (K1) normal, (K2) dry, and (K3) extremely dry scenarios. (K1-n,K2-n and K3-n represents the nth class of solutions in the normal, dry and extremely dry scenario separately, n=1~6.)＂

＂Fig. 7. Water supply from each reservoir to connected water works in each month in the normal scenario $10^4 m^3$

(K1-n represents the $n^{th}$ class of solutions in the normal scenario, n=1~6.)＂

＂Fig. 8. Pareto fronts of the second layer in the PTSOA model after 500 simulations with the hierarchical optimal algorithm in the normal, dry and extremely dry scenarios. (F1 represents the total amount of water retained in water works ,$10^4 m^3$; F2 represents the amount of unconventional water supplied,$10^4 m^3$. The direction of optimization is

from the top-right corner to the bottom-left corner. K1-n represents the nth class of solutions in the normal scenario, K2-n represents the nth class of solutions in the dry scenario, and K3-n represents the nth class of solutions in the extremely dry scenario, n=1~6.)"

"Fig. 9. Illustration of parallel-reference Pareto sets from the third layer in the PTSPOA model attained across all runs for the (S1) normal, (S2) dry, and (S3) extremely dry scenarios (S1-1 represents the normal scenario with the minimum total amount of water retained in water works, S1-2 represents the normal scenario with the maximum unconventional water supply and S1-3 represents the normal scenario with the maximum synergy degree in the second layer)"

"Fig.10. Comprehensive benefit in five sub-regions after the regional collaborative allocation of water resources (S1 represents normal scenario, S2 represents dry scenario, and S3 represents extremely dry scenarios; S1-1 represents the normal scenario with the minimum total amount of water retained in water works, S1-2 represents the normal scenario with the maximum unconventional water supply and S1-3 represents the normal scenario with the maximum synergy degree in the second layer)"

*10.Line 251, ...*

*Each section that describes the three layers of the process shares the same subsection name; I would recommend renaming them to avoid any confusion.*

Thanks for your comment! The names of the sub-sections have been corrected in the revised version. For example: 2.1.1Objective functions of the first layer; 2.1.2 Constraints of the first layer; 2.2.1 Objective functions of the second layer; 2.2.2 Constraints of the second layer; 2.3.1 Objective function of the third layer; 2.3.2 Constraints of the third layer.

*11. Line 228: space after "interactions"*

Thanks for your comment! The whole paper has been checked and the missing spaces have been added like the space after "interactions".